# Chemoenzymatic synthesis of genetically-encoded multivalent liquid *N*-glycan arrays

Chih-Lan Lin[1], Mirat Sojitra [1], Eric J. Carpenter[1], Ellen S. Hayhoe[1], Susmita Sarkar[1], Elizabeth A. Volker[1], Chao Wang[2], Duong T. Bui [1], Loretta Yang[3], John S. Klassen[1], Peng Wu [2], Matthew S. Macauley [1,4], Todd L. Lowary [1,5,6] & Ratmir Derda [1] ✉

Cellular glycosylation is characterized by chemical complexity and heterogeneity, which is challenging to reproduce synthetically. Here we show chemoenzymatic synthesis on phage to produce a genetically-encoded liquid glycan array (LiGA) of complex type *N*-glycans. Implementing the approach involved by ligating an azide-containing sialylglycosyl-asparagine to phage functionalized with 50–1000 copies of dibenzocyclooctyne. The resulting intermediate can be trimmed by glycosidases and extended by glycosyltransferases yielding a phage library with different *N*-glycans. Post-reaction analysis by MALDI-TOF MS allows rigorous characterization of *N*-glycan structure and mean density, which are both encoded in the phage DNA. Use of this LiGA with fifteen glycan-binding proteins, including CD22 or DC-SIGN on cells, reveals optimal structure/density combinations for recognition. Injection of the LiGA into mice identifies glycoconjugates with structures and avidity necessary for enrichment in specific organs. This work provides a quantitative evaluation of the interaction of complex *N*-glycans with GBPs in vitro and in vivo.

The surface of every cell is coated with complex glycans, installed on lipids or as post-translational modifications on proteins, forming a glycocalyx at which cellular interactions occur[1–4]. Cell-surface glycans mediate biological processes as diverse as cell–cell adhesion, bacterial and viral infection, and immune regulation[5–9]. Interfering with these processes is a demonstrated strategy for drug action[10–13]. Understanding the recognition properties of glycan-binding proteins (GBPs) is necessary to unravel the role of glycans in cellular interactions and function[14–17]. A property of particular interest is density, which plays an essential role in glycan–GBP interactions[18–21]. Most GBPs have multiple glycan-binding sites, each specifically recognizing a glycan moiety or a fragment of a complex glycan[14,21,22]. The affinity between monovalent glycans and a GBP binding site are typically in the low $\mu$M range. For enhancement of weaker interactions, the spatial organization of

multiple copies of the same glycan on the cell surface both influences the initial recognition event and binding and generates clustered saccharide patches that enhance avidity via multivalency[21,23,24].

Previous efforts to probe the effect of glycan structure and density on GBP-recognition, have used microarrays[25–27] in which glycans are displayed on (usually) glass slides that, in principle, mimic their natural valency and spatial presentation[28]. Despite their enormous utility, conventional glycan arrays have drawbacks; for example, they cannot assay interactions with intact cells. Furthermore, with the exception of BSA-conjugate arrays[19,29,30], it is challenging to systematically probe the effect of density using solid-phase arrays. The synthesis of homogeneous multivalent displays of glycans on polymers, dendrimers, liposomes, and other carriers offers a higher level of density control[31–35]. Many such displays have been used to study

[1]Department of Chemistry, University of Alberta, Edmonton AB T6G 2G2, Canada. [2]Department of Molecular Medicine, The Scripps Research Institute, 10550 N. Torrey Pines Road, La Jolla, CA 92037, USA. [3]Lectenz Bio, 111 Riverbend Rd, Athens, GA 30602, USA. [4]Department of Medical Microbiology and Immunology, University of Alberta, Edmonton, AB T6G 2E1, Canada. [5]Institute of Biological Chemistry, Academia Sinica, Taipei, Taiwan. [6]Institute of Biochemical Sciences, National Taiwan University, Taipei, Taiwan. ✉e-mail: ratmir@ualberta.ca

cellular responses[20,35–37]. Unlike arrays, these multivalent displays cannot encode multiple types of glycans and, as a result, can probe only limited numbers of structures. To bridge these technologies, we recently developed liquid glycan arrays (LiGAs)[38], which employ phage virions with DNA barcodes in the phage genome as a display platform. These genetically-encoded libraries of structurally-diverse multivalent glycoconjugates are powerful probes of GBP–glycan interactions[38]. Compared to monovalent DNA-coded glycan arrays[39–42], the LiGA strategy provides the ability to control and encode glycan density. LiGA can also probe interactions with cells both in in vitro and in vivo. In our initial report describing the LiGA technology[38], we used relatively simple glycan structures, produced through chemical or chemoenzymatic synthesis prior to their incorporation onto M13 phage. Recent advances in the isolation of *N*-glycans from natural sources[43,44] and chemoenzymatic synthesis[45,46] provides an attractive opportunity for building custom *N*-glycan LiGAs directly on phage.

The Flitsch group has employed biocatalysis for on-DNA glycan synthesis[40]. This pioneering study constructed carbohydrate-based libraries using enzymatic oxidation and/or glycosylation. The glycans generated were only monovalent; nevertheless, this report is an important precedent for chemoenzymatic glycan array synthesis. Later, the Cha and Reichardt groups independently used chemoenzymatic methods to construct multivalent displays of glycans on glass slides[47,48]. Characterizing the completion of the enzymatic reactions was difficult and the authors inferred that most did not reach full conversion, even when performed under conditions that yielded full conversion of analogous glycans in solution. Most recently, independent reports from Wu and Capicciotti describe enzymatic remodeling of glycans on cells[49–52]; the major bottleneck in such approaches is reaction characterization. Introduction of a new monosaccharide (e.g., Neu5Ac) on the cell surface is detected by increased binding of the cell to a lectin. However, direct detection of changes directly in cellular glycan composition by mass spectrometry is challenging[53]. The challenges present in state-of-the-art enzymatic glycosylation on-cells, on-glass, and on-DNA prompted us to develop chemoenzymatic glycan synthesis "on-phage" with a complimentary analytical method to monitor reaction conversion. We anticipated that this approach would enable preparing genetically-encoded multivalent displays of glycans with defined structure and, importantly, quantitatively-defined densities.

We describe here the successful implementation of this strategy, which has provided new LiGA components—bacteriophages equipped with DNA-barcodes displaying *N*-glycans. We also observed that on-phage enzymatic conversion in solution occurred more efficiently than on a two-dimensional (glass slide) display. Access to this library has, in turn, allowed us to study glycan–GBPs interactions in vitro, on cell surfaces, and in mice, with a particular focus on understanding the impact of glycan density on recognition.

## Results

### Chemoenzymatic synthesis of glycan-phage conjugates

Previously[38], we assembled a library of glycan-coated phages using strain-promoted azide–alkyne cycloaddition (SPAAC) to ligate oligosaccharides with alkyl-azido linkers to dibenzocyclooctyne (DBCO)-modified M13 phage, each containing a distinct DNA barcode. We chose to employ this approach again to prepare an *N*-glycan library using a heterogenous sialylglycopeptide (SGP, **1**, Fig. 1a, Supplementary Fig. 1-7) from egg yolk[43,44], a commonly employed starting material for chemoenzymatic *N*-glycan preparation[45].

Initially, we used a published route[44] to trim **1** to a homogenous *N*-glycan and ligate it to phage. To do this, **1** was treated with pronase to provide a mixture of asparagine (Asn)-linked biantennary oligosaccharides **2** (Fig. 1a, Supplementary Fig. 2). Subsequent treatment of **2** with neuraminidase and then β-galactosidase resulted in homogeneous GlcNAc-terminating biantennary structure, **4** (Supplementary

Fig. 3-4), which was *N*-acylated on the amine of Asn with 8-azido-octanoic acid NHS-ester **5**[54] to yield **6** (Supplementary Fig. 5). Biantennary glycan **6** retained its natural *N*-linkage to Asn, whereas the azido-linker allowed ligation to DBCO- modified M13 phage by SPAAC. Monitoring the SPAAC reaction by MALDI-TOF MS (Fig. 1c) showed that ligation of *N*-glycan **6** required longer times (24 h) compared to the 1–2 h reaction times needed for smaller glycans[38]. We then used similar steps to install a heterogeneous glycosyl asparagine derivative. Thus, Asn-linked *N*-glycans **2** were acylated with **5** to yield a mixture of *N*-glycans **7**, which was ligated to DBCO-modified M13 phage (Fig. 2a). MALDI-TOF MS confirmed the modification (Fig. 2c, d and see Supplementary Fig. 8 for optimization of MALDI conditions). Peaks S1 and S2 correspond to natural symmetric and asymmetric biantennary structures. Peak S2' represents the cleavage of one sialic acid from a symmetric glycan during MALDI-TOF MS detection as confirmed by enzymatic treatment described below.

### On-phage enzymatic reactions with multivalent displayed of *N*-glycans

Phages decorated with either homogeneous or heterogeneous glycans can be used for chemoenzymatic glycan modification. Such on-phage trimming or elongation of glycans facilitates the preparation of glycoconjugates with consistent densities across a range of structures. MALDI-TOF MS confirmed that β-galactosidase treatment quantitatively removes terminal galactose residues from glycans on phage (Supplementary Fig. 9). Similarly, neuraminidase trimming of sialic acids in phage-displayed SGP yielded *N*-glycans with either one or two terminal galactose residues (peaks P1 and P2, Fig. 2e). Subsequent β-galactosidase treatment (Fig. 3a) revealed progressive cleavage of both galactose residues with complete disappearance of the symmetric structure after ~2 h (Fig. 3c), transient accumulation of mono-galactosylated glycans I1, and then their disappearance after ~4 h (Fig. 3c). Tandem neuraminidase and β-galactosidase treatment of heterogeneous **7** on phage, thus, quantitatively gave a homogeneous glycosylated product (Fig. 3a). In contrast, direct β-galactosidase treatment of **7** on phage yielded no observable changes (Supplementary Fig. 10), confirming that the glycan contains no species with terminal galactose residues and that the P2' peak observed by MALDI-TOF MS are indeed "ghost" species generated by sialic acid cleavage during analysis (Supplementary Fig. 10). Further evidence comes from model 6'SLN (α-Neu5Ac-(2→6)-LacNAc) glycans ligated to phage, which can be cleaved by β-galactosidase; such cleavage was blocked by sialylation of the galactose residues (Supplementary Fig. 11). The homogenous biantennary glycan with terminal *N*-acetylglucosamine (GlcNAc) was further treated with β-*N*-acetylglucosaminidase to cleave the GlcNAc residues yielding a homogenous *N*-glycan structure with terminal mannose (paucimannose) (Supplementary Fig. 12). These results confirm that efficient multi-step enzymatic trimming of *N*-glycans on phage is possible.

We also explored on-phage *N*-glycan synthesis using glycosyltransferases. In model studies, phages with *N*-glycans terminating with GlcNAc on phage (Supplementary Fig. 13) or in solution (Supplementary Fig. 7) were treated with β-(1→4)-galactosyltransferase (B4GalT1) and uridine 5'-diphosphogalactose (UDP-Gal)[45] to give, after ~40 h, phages with lactosamine (LacNAc)-terminating structures (Supplementary Fig. 13b). We also applied these conditions to transfer galactose to heterogeneous asialo-SGP on phage (Supplementary Fig. 14), to provide a homogenous biantennary *N*-glycan with terminal galactose residues. MALDI-TOF MS confirmed a time-resolved conversion of peak P1 (asymmetric glycan) to species P2 (symmetric glycan) over the course of 8 h (Supplementary Fig. 14b). In model studies of sialylation, quantitative addition of sialic acid to LacNAc- and lactose-phages was achieved using recombinant α-(2→6)-sialyltransferase from *Photobacterium damselae* (Pd26ST) and cytidine-5'-monophosphate-*N*-acetylneuraminic acid (CMP-Neu5Ac). After

9 h, α-Neu5Ac-(2→6)-LacNAc and α-Neu5Ac-(2→6)-Lac-phages were formed quantitatively (Supplementary Fig. 11b and 15b). Pd26ST also transferred a sialic acid derivative bearing a 3-butynamide group at C-5[55,56] to Lac-phage (Supplementary Fig. 15c). This alkyne handle can be exploited for further chemical derivatization of phage-displayed glycans. We further demonstrated that α-(2→3)-sialyltransferase (Pm2,3ST) can be used on model LacNAc glycans displayed on phage (Supplementary Fig. 16) and Gal-terminated N-glycans displayed on phage (Supplementary Fig. 17a, c).

The loss of sialic acid during the MS analysis made it difficult to confirm reaction completion by this technique alone (Supplementary Fig. 11b and 15b). After the sialylation, the phage was therefore treated with β-galactosidase to cleave the terminal galactose in any unreacted LacNAc or Lac moieties (Supplementary Fig. 11b and 15g). Using this approach, we confirmed that sialylation proceeds to completion and that asialoglycans observed in MALDI-TOF MS are "ghost" peaks. If necessary, tandem cleavage of Gal and then GlcNAc can also distinguish partially and fully sialylated glycans (Supplementary Fig. 18). We also observed a reduction of MALDI-TOF MS signal intensity upon sialylation. To ensure that this decrease is not due to glycan degradation during the enzymatic reaction, we performed a tandem

Pd26ST-catalyzed sialylation and neuraminidase de-sialylation. The intensity for the LacNAc-pVIII conjugate in the mass spectrum was similar before and after the sialylation/de-sialylation cycle (Supplementary Fig. 19) confirming that the decrease in intensity is due to decreased ionization capacity[57]. Using these optimized synthesis and monitoring procedures, we performed a two-step on-phage enzymatic extension using B4GalT1 and Pd26ST to yield a symmetric biantennary sialylated N-glycan (Fig. 4). Phage-bound **7** was first extended by B4GalT1 and UDP-Gal (Fig. 4b and d). After purification by PEG-precipitation, Pd26ST-catalyzed transfer of Neu5Ac yielded a homogeneous product P2 on phage (Fig. 4b and d). Modification with both B4GalT1 and Pd26ST did not proceed as efficiently for densely glycosylated phages (1000 glycans per phage) as they did for phages with medium density glycosylation (750 or less). Specifically, galactosylation of phage containing 1000 copies of glycan by B4GalT1 required 7 days for completion, whereas modification of 750 glycans/phage by the same enzyme was done in 1 day (Supplementary Fig. 20). Similarly, modification by Pd26ST was slower at 1000 glycans/phage than 750 or lower densities (Supplementary Fig. 21). Similar trends were observed for modification by Pd26ST. (Supplementary Fig. 21) Such observations resemble observations from Fitch, Webb, and co-workers that

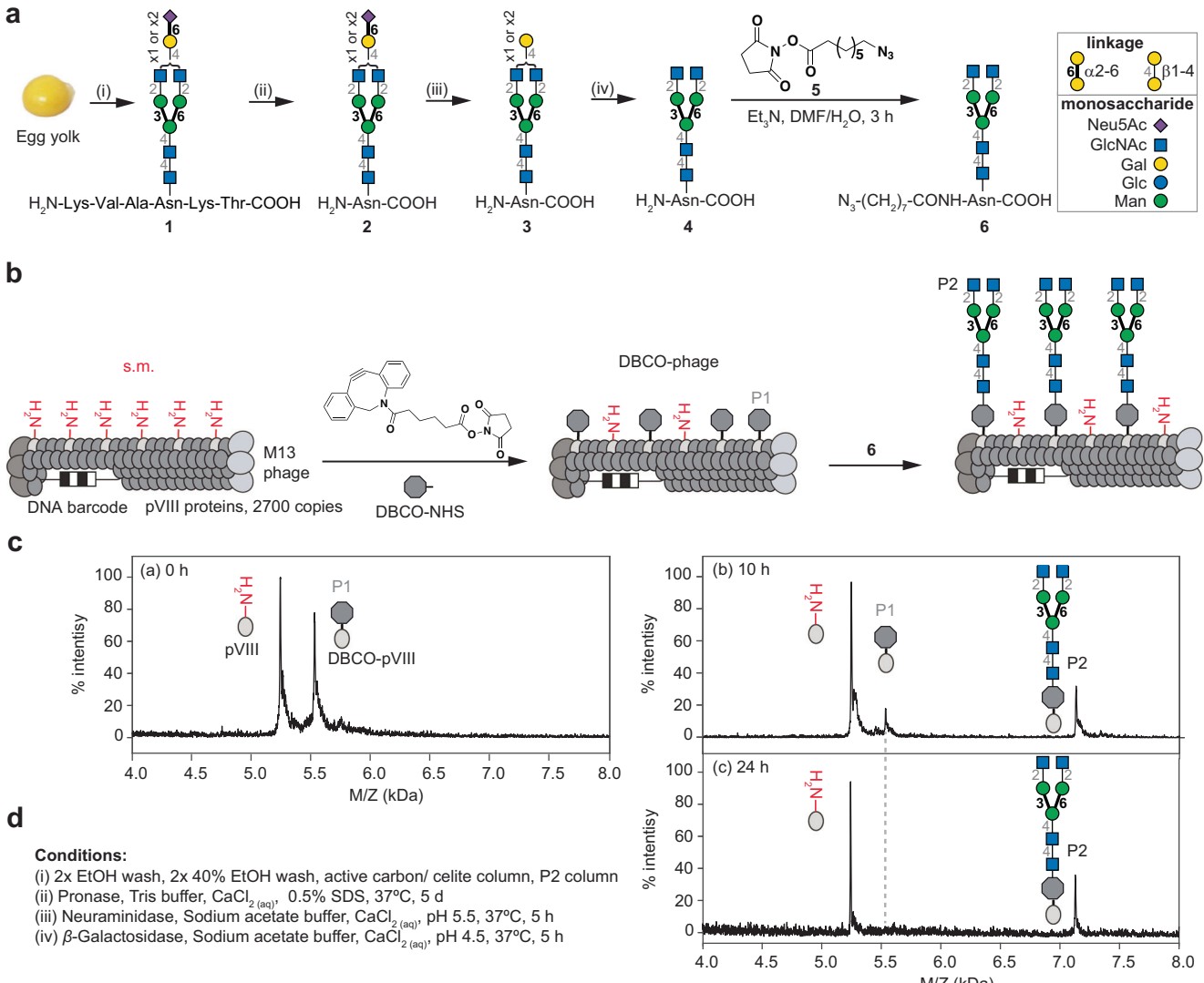

**Fig. 1 | Synthesis and characterization of LiGA components. a** Isolation of SGP from egg yolk and trimming steps to afford homogeneous azido-functionalized N-glycan **6**. **b** Representation of two-step linkage of N-glycan to phage. **c** MALDI-TOF MS characterization of starting material (pVIII protein), alkyne-functionalized product (DBCO-pVIII, P1), and glycoconjugate product (P2). **d** Conditions of each step to afford product **6**.

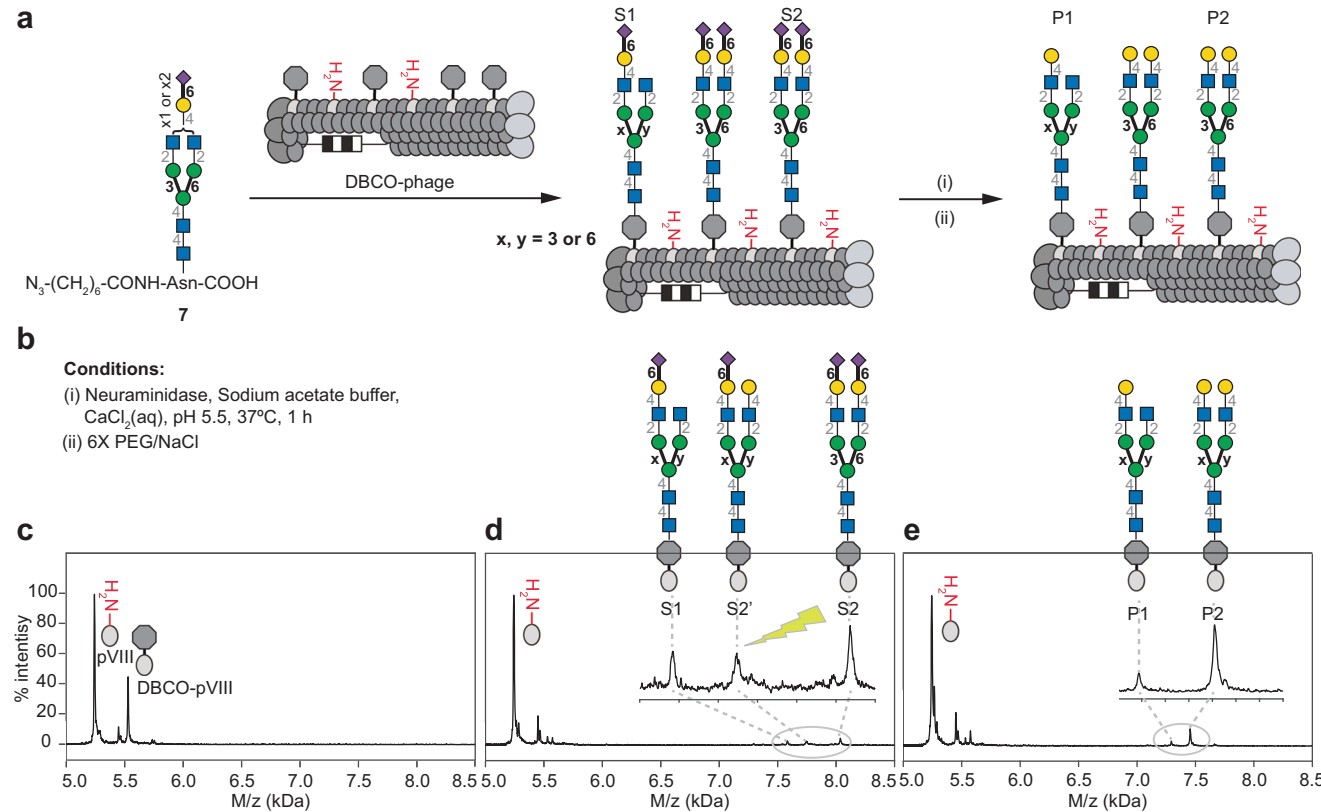

**Fig. 2 | On-phage enzymatic trimming of sialic acid using neuraminidase.**
**a** Preparation and attachment of azido-modified Asn-linked biantennary SGP **7** to M13 phage. **b** neuraminidase treatment to afford mixtures of conjugates with either one or two terminal galactose residues on pVIII. **c** MALDI-TOF MS characterization of conjugation of DBCO-NHS to pVIII. **d** S1 and S2 products of ligation of **7**, and S2′, a MALDI-TOF MS artifact. **e** P1 and P2 phage-displayed glycans generated by neuraminidase treatment.

demonstrated a dramatic reduction of the rate of enzymatic modification of glycans on high vs. low density liposomes[58]. The combined results confirmed that multi-step enzymatic glycan remodeling can be used to create *N*-glycans directly on phage, but that the modification rates slow down near a density of 1000 glycans per 0.7 micron-long phage. To evaluate further the scope of enzymatic modification, we used *H. pylori* α-(1→3)-fucosyltransference (Hp1,3FT) (Supplementary Fig. S22, 23). Similar to reported observations[59], Hp1,3FT rapidly modified Gal-terminated *N*-glycans on phage (Supplementary Fig. 22) but not α-(2→6)-sialic acid-terminated *N*-glycans (Supplementary Fig. 23). The latter glycan can be modified after neuraminidase treatment (Supplementary Fig. 23). Both glycans exhibited nearly 100 times stronger binding to Aleuria aurantia lectin (AAL) when compared to the phage displaying a precursor glycan (Supplementary Fig. 24).

**Binding of LiGA to glycan-binding proteins in vitro**
Developing a robust method to synthesize phage-displayed *N*-glycans made it possible to study the effect of *N*-glycan structure and density on GBP binding. To study the effect of density, we synthesized a library of six *N*-glycans displayed at five different mean densities (50, 150, 500, 750 and 1000 glycans/phage) (Supplementary Fig. 20, 21, and 25–28). An example of our ability to control glycan density is shown in Supplementary Fig. 29. The mean density was set by installing a range of 50–1000 copies of DBCO per phage (confirmed by MALDI-TOF MS), followed by complete conjugation with **7** and, finally, quantitative chemoenzymatic conversion of phage-SGP to the desired structures (again confirmed by MS). The resulting library, dubbed "LiGA6×5", was used to analyze binding to fifteen lectins: *Sambucus nigra*-I (SNA-I), Concanavalin A (ConA), *Ricinus communis* agglutinin (RCA-I), *Lens culinaris* hemagglutinin (LCA), *Pisum sativum* agglutinin (PSA),

*Galanthus nivalis* lectin (GNL), *Erythrina cristagalli* agglutinin (ECL), Wheat Germ agglutinin (WGA), CD22, Aleuria aurantia lectin (AAL), Asialoglycoprotein receptor (ASGPR), Carbohydrate-binding modules 40 (diCBM40), SiaFind™ α-(2→6)-specific reagent, SiaFind™ Pan-specific Lectenz (Fig. 5 and Supplementary Fig. 29–40) as well as cells overexpressing CD22 and DC-SIGN (dendritic cell-specific intercellular adhesion molecule-3-grabbing non-integrin) (Fig. 5). In these experiments, LiGA6×5 was incubated in wells coated with each lectin before unbound phages were removed by washing. Bound particles were eluted with 1 M HCl and analyzed by next generation sequencing (NGS). As a metric, we used the fold change (FC) difference[38] in copy number of each phage with respect to its copy in LiGA6×5 incubated in BSA-coated wells that underwent analogous processing (wash, elution, NGS).

SNA-I lectin[60] and CD22 (Siglec-2) both recognize α-(2→6)-sialylated *N*-glycans and, as expected, we observed interaction of only α-(2→6)-sialylated components of LiGA with these proteins (Fig. 5d–f and Supplementary Fig. 29, 30). SNA-I bound strongly to a medium density of glycans (~150 glycans per phage), whereas phage with ≤50 or ≥500 glycans bound significantly lower. In contrast, CD22 required ≥500 glycans per phage for significant binding (Fig. 5e). The same >500 glycans per phage requirement for binding was seen in a more biologically-relevant environment: CD22 expressed on the surface of cells (Fig. 5f). Asialoglycans exhibited no statistically significant binding to SNA-I and CD22 proteins or CD22⁺ cells at any density. We previously confirmed that decrease of lectin–glycan interactions at high glycan density occurs due to steric occlusion[38,61], here we tested this again by making a phage that displayed a mixture of sialylated and non-sialylated glycans (Supplementary Fig. 41). The mixtures of functional and non-functional glycans did not bind to SNA when the total

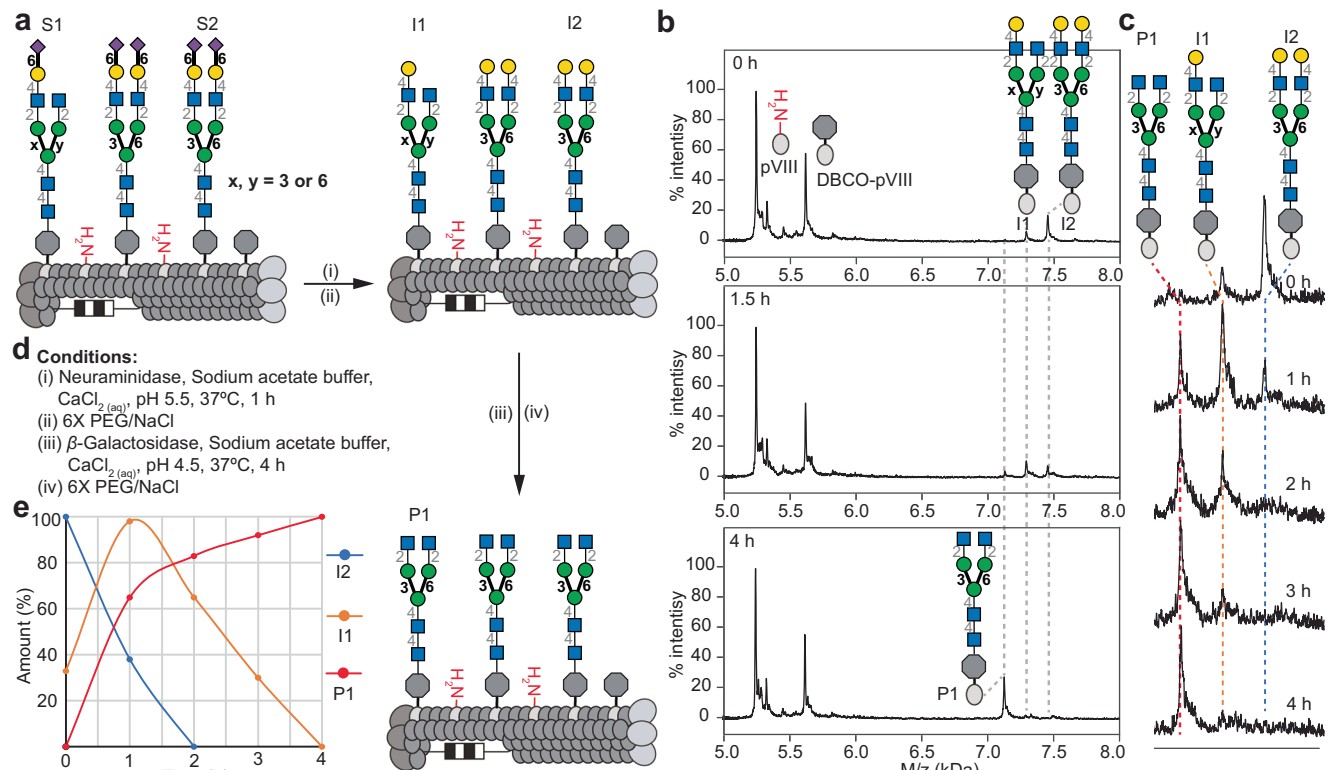

**Fig. 3 | On-phage two-step enzymatic trimming of biantennary *N*-glycans by neuraminidase and β-galactosidase. a** Biantennary N-glycans S1, S2 were cleaved by neuraminidase to yield galactose-terminated intermediates I1 and I2. Subsequent treatment with *β*-galactosidase yielded a homogeneous terminal GlcNAc heptasaccharide product P1 on pVIII. **b** MALDI-TOF MS characterization after *β*-galactosidase treatment. **c** Progress of β-galactosidase treatment as monitored by MALDI-TOF MS over time. **d** Conditions of each glycosidase treatment. **e** Plot of time course results with the amount (%) of I1, I2 and P1 over time (h).

density of glycans reached 1000, even when density of sialoglycans was low in such mixture (Supplementary Fig. 42). We are not aware of any reports describing non-overlapping density dependence of SNA-I and CD22 proteins, which may stem from differences in the accessibility of protein clusters on the plate or cell surface, or 50–100x weaker affinity of interaction of sialylated glycans with CD22 when compared to SNA (75 μM and 0.77 μM respectively)[62,63] (Supplementary Fig. 43, 44). To provide additional support for this affinity hypothesis, we tested a dimeric SiaFind™ α-(2→6)-specific protein, which binds to α-(2→6)-sialic acid with micromolar $K_d$ and monomeric SiaFind™ Pan-specific Lectenz®, which binds both α-(2→6) and α-(2→3)-sialic acid with 0.2–0.4 μM affinity. Unlike tetrameric SNA lectin, both SiaFind™ reagents preferred α-(2→6)-sialoglycans at higher density (Supplementary Figs. 31c and 32c). This experiment suggests that density-dependence might emanate from oligomerization state rather than affinity. Enzymes can be used to remodel not only individual glycans but also their mixtures (libraries). To show this capacity, we remodeled LiGA6×5 using α-(2→3)-sialyltransferase (Pm2,3ST) and showed that the glycans in the remodeled library bind to diCBM40 (Supplementary Fig. 33e), SiaFind™ Pan-specific Lectenz® and SiaFind™ α-(2→6)-specific reagent (Supplementary Figs. 31e and 32e). We observed that SiaFind™ Pan-specific Lectenz® binds to the newly installed α-(2→3)-sialosides but SiaFind™ α-(2→6)-specific reagent does not. Again, SiaFind™ Pan-specific Lectenz® preferred α-(2→3)-sialosides only at high density. SiaFind™ Pan-specific Lectenz® and SiaFind™ α-(2→6)-specific reagent are monomeric and dimeric, respectively, whereas SNA is tetrameric and the observed density preferences might stem from oligomerization state of lectin rather than $K_d$ which is approximately micromolar for all. An interplay of structure and density was also observed in a mannose binding lectin family (ConA, LCA, PSA, GNL and DC-SIGN). ConA recognizes a wide range of biantennary *N*-glycans

and tolerates multiple extensions[60], and LiGA6×5 detected binding of ConA to nearly all biantennary *N*-glycans on phage with the exception of paucimannose with terminal GlcNAc (Fig. 5 and Supplementary Fig. 34). Binding occurred at medium density, 150–750 copies per phage, depending on the glycan, but at 1000 glycans per phage there was no detectable binding to any *N*-glycan. To test the significance of density-dependent binding, we produced a library in which every density of paucimannose was linked to seven distinct SDB (DNA barcodes) (Supplementary Fig. 45). This multi-SDB (MSDB encoding of five densities of Man₃) confirmed that phage displaying 150 paucimannose *N*-glycans bound to ConA significantly better than phage that display 50 or 500 copies of the same glycan. Furthermore, binding to 1000 copies of paucimannose per phage is significantly weaker than any other construct (Supplementary Fig. 46). Lack of ConA binding to paucimannose with terminal GlcNAc is not in agreement with earlier observations by Cummings and coworkers (Supplementary Fig. 34c)[28] and the Consortium for Functional Glycomics (CFG) array (Supplementary Fig. 34e). We observed no binding at any glycan density (Supplementary Fig. 47a) and to rule out the possibility of competition between glycans, we prepared simple mixtures (Supplementary Fig. 47b) that contain only paucimannose with GlcNAc **6**. Again, the phage clones that displayed **6** showed no binding. However, when the LiGA was treated with β-N-acetylglucosaminidase S, the non-binding clones that used to carry glycan **6** exhibited strong binding, due to removal of GlcNAc (Supplementary Fig. 46e) while binding of other glycans to ConA was not affected. This experiment showed that enzymes can be used to remodel not only individual glycans but also their mixtures (libraries).

Unlike ConA, the mannose binding lectins LCA and PSA bound to the core Man₃ epitope in all six *N*-glycans (Fig. 5h and Supplementary Fig. 35). Some glycan array studies have suggested that core

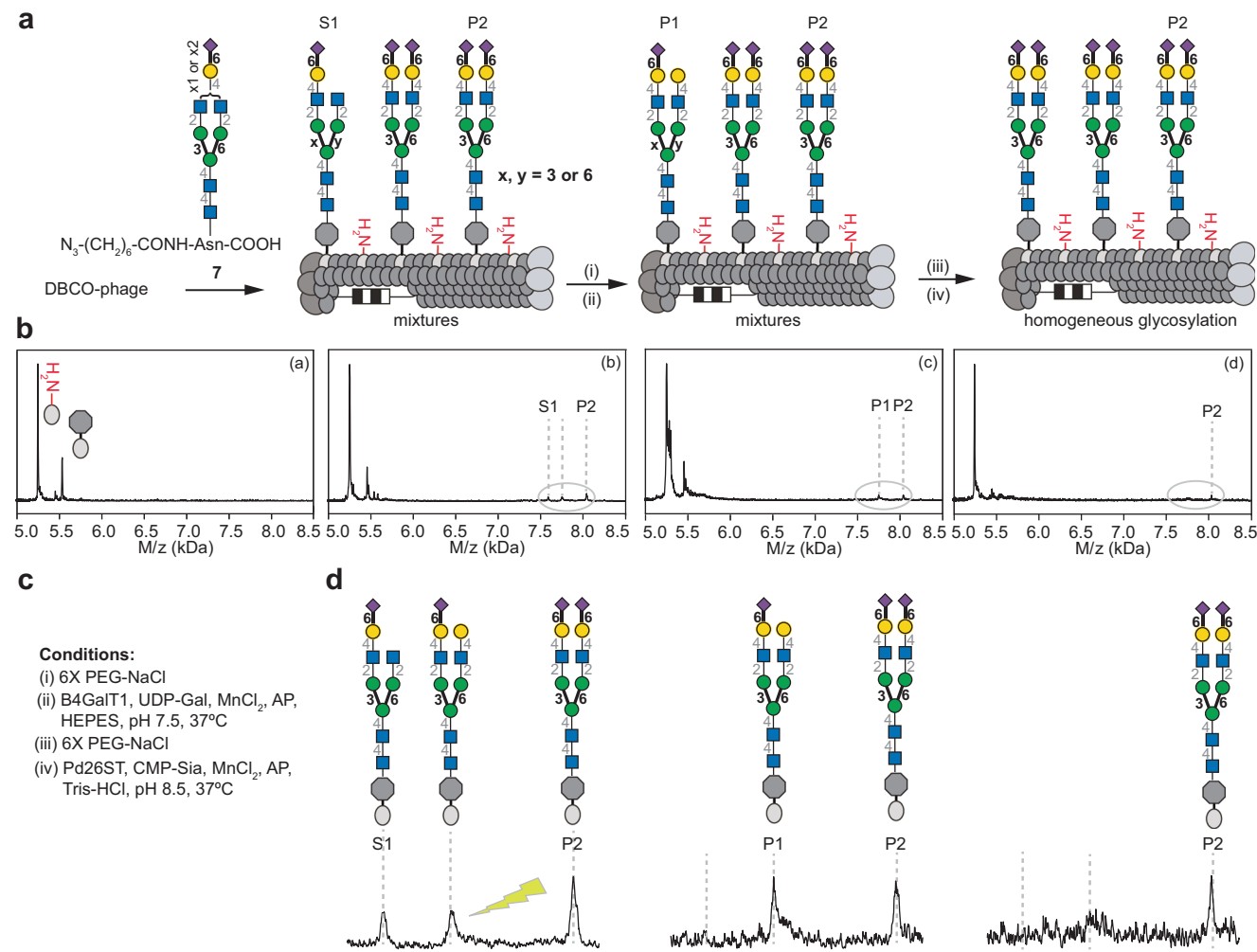

**Fig. 4 | On-phage two-step enzymatic extension of bi-antennary *N*-glycan by B4GalT1 and Pd26ST. a** Sequence of β-(1→4)-galactosylation and α-(2→6)-linked sialylation to afford homogeneous sialylated product P2. **b** MALDI-TOF MS: (**b**) substrates S1, S2 were products of the ligation reaction. (as described in Fig. 2) (**c**) B4GalT1-catalyzed galactosylation converted S1 to P1. **d** Pd26ST-catalyzed sialylation converted P1 to P2 yielding a homogeneous display of P2. **c** Conditions for enzymatic reactions. **d** Detailed changes of each peak on MALDI-TOF MS. The lightning symbol indicate that these peaks are "ghost" peaks that do not respond to β-galactosidase treatment.

fucosylation is required for LCA or PSA to recognize the Man₃ epitope[28] (Supplementary Figs. 35d, 36d) whereas other investigations[60] observed binding without fucose (Supplementary Fig. 35e, 36e). Affinities of Man₃ for LCA ($K_d = 10\,\mu M$) and PSA ($K_d = 15\,\mu M$) have also been measured by calorimetry[64]. Notably, PSA recognized α-(2→6)-sialylated glycans at 1000 glycans/phage whereas binding to asialoglycans at 1000 glycan/phage density was significantly decreased ($n = 5$ independent experiments). A possible explanation for this observation is a change the conformation/accessibility of Man₃ with and without the negatively charged sialic acid epitopes. Finally, the GNL lectin, which is known to recognize paucimannose, bound strongly to phages that display intermediate copy number of this structure (150–750 copies, Supplementary Fig. 37b) and more weakly to any other *N*-glycans (Fig. 5h, Supplementary Fig. 37). A detailed study of GNL:Man₃ recognition using MSDB-Man3 library described above confirmed this observation (Supplementary Fig. 48).

Recognition of LiGA6×5 by DC-SIGN was dramatically different from ConA, LCA, PSA or GNL. Using DC-SIGN⁺ rat fibroblasts, we observed that DC-SIGN does not tolerate terminal LacNAc or sialyl-LacNAc on the core Man₃, regardless of glycan density (Figs. 5g, h). In line with our previous report[38], a narrow range of density – 500 Man₃

epitopes/phage—was optimal for interaction with DC-SIGN⁺ cells. This preference shifted to higher density for GlcNAc-terminated Man₃, corroborating an earlier study[65] that showed that DC-SIGN recognition requires a high density of GlcNAc-terminated Man₃ epitopes[65]. In agreement with our previous report, the binding to Man₃ was ablated at ≥1000 glycans/phage due to steric occlusion of tightly packed Man₃ epitopes[38].

In the Lac/LacNAc binding lectin family, RCA-I lectin[60] bound to LiGA6×5 components decorated with galactose-terminated *N*-glycans and those with α-(2→6)-linked sialic acid (Fig. 5h and Supplementary Fig. 38). LiGA measurements matched prior observations[28] and uncovered a previously unknown bimodal density dependence of RCA-I binding (Supplementary Fig. 38). ECL, which is known to recognize terminal β-(1→4)-linked galactose[60], bound selectively at low concentration of lectin ([ECL] = 10 μg/mL) to phage displaying terminal galactose. At high concentration ([ECL] = 20 μg/mL), this lectin also recognized phages with α-(2→6)-sialylated galactose (Fig. 5h and Supplementary Fig. 39). WGA, which binds to various terminal[60] and internal GlcNAc[66], exhibited binding to all LiGA components including Man₃GlcNAc₂ with internal GlcNAc (Fig. 6c, Supplementary Fig. 40). We further tested the significance of density dependent binding of WGA to Man₃ by using MSDB-Man₃ (Supplementary Fig. 49). We

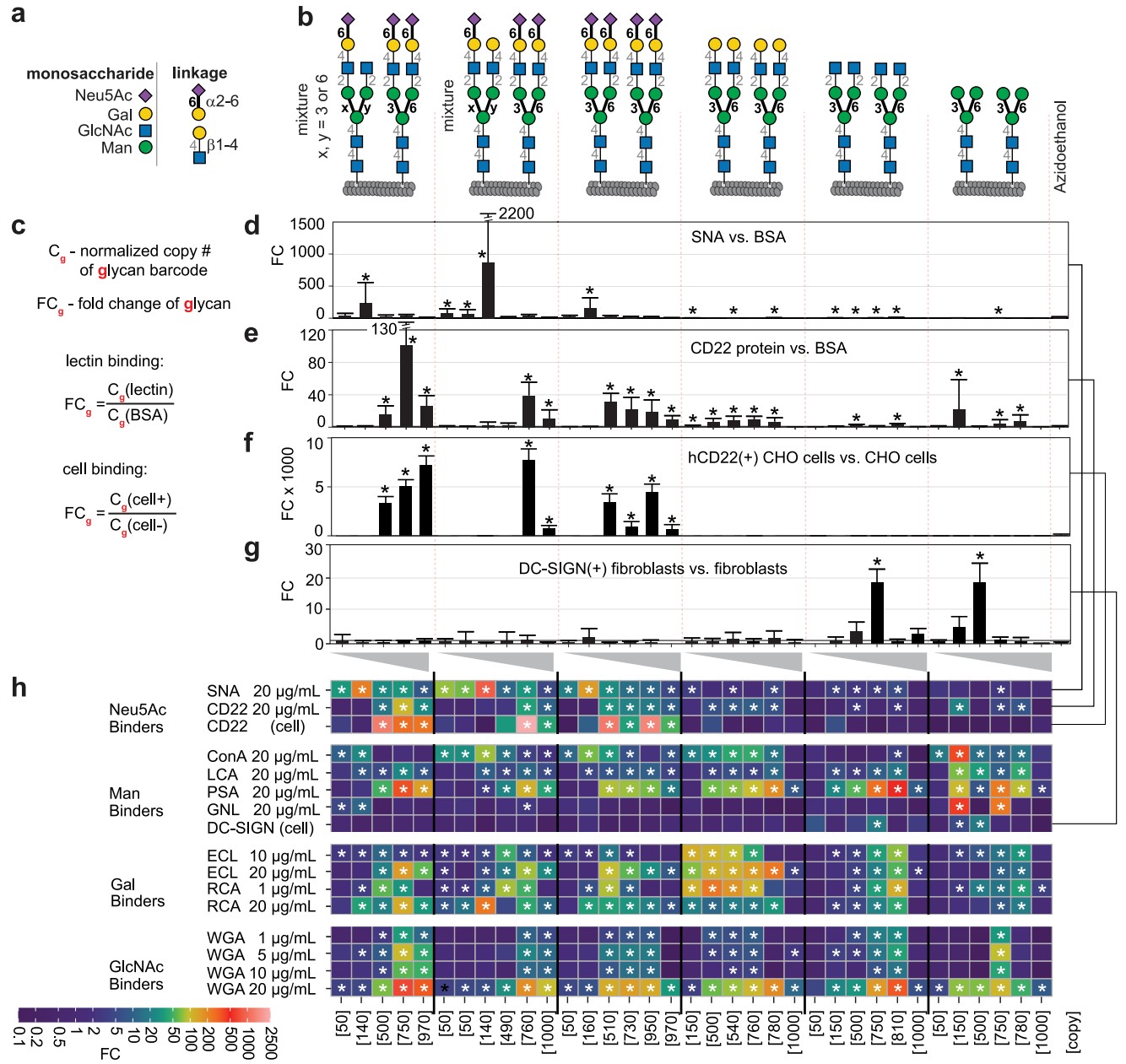

**Fig. 5 | Interaction of LiGA6×5 with SNA, CD22 and cells overexpressing CD22 or DC-SIGN measure avidity-based responses. a** Glycan abbreviations. **b** LiGA6×5 is composed of six *N*-glycans each at five display densities. Two of the six are mixtures. **c–g** Bars show the best-fit fold change (FC) calculated by edgeR, with normalization by assuming equality of azidoethanol labeled phages; error bars represent standard deviation propagated from the variance of the normalized sequencing data; multiple test control was by false discovery rates (FDR) from Benjamini-Hochberg adjustment; **d, e** Binding of LiGA6×5 to wells coated with SNA-I (**d**) or CD22 protein (**e**). **f, g** Binding of LiGA6×5 to CD22 lectin overexpressed on CHO cells (**d**) and DC-SIGN expressed on rat fibroblasts (**e**); normalization in (**f**) was done using Trimmed Mean of M-values (TMM) due to the low abundance of reference reads in this dataset. **h** Heatmap representation of the binding of 10 purified lectins and 2 cell-displayed lectins to LiGA6×5. Data from **d–g** (SNA, CD22, CD22⁺ and DC-SIGN⁺ cells) is repeated in the heatmap for consistency. In **d–h**, asterisks (*) indicate FDR ≤ 0.05 from independent experiments (*n* = 6 LCA, *n* = 8 ConA, *n* = 11 WGA 20 µg/mL, *n* = 2 CD22-cell +/-, *n* = 13 BSA, *n* = 5 all other cases).

observed binding of WGA to MSDB-Man₃ at densities from 500 to 1000 glycans per phage. No binding occurred at densities of 50 and 150 copies of paucimannose. Binding to specific *N*-glycans was dictated by density of the glycan on phage and attenuated by the concentration of WGA used to produce the WGA-modified surface (the surface density of WGA) (Supplementary Fig. 40). The recognition preferences of ECL and WGA aligned with some earlier glycan array experiments[60] (Supplementary Fig. 39c, 39e, 40c, and e) but diverged from others (Supplementary Figs. 39d, 40d)[28,66]. In these latter cases, the binding of ECL or WGA to sialylated-*N*-glycans was not observed

and we note that these lectins bound to sialylated structures in LiGA6×5 only at high density (750–1000 glycans/phage). In contrast, binding of asialoglycans to ECL and WGA was bimodal; binding was optimal at intermediate glycan densities. We tested whether binding to ECL and WGA at high densities of glycans decreased due to steric occlusion (Supplementary Fig. 42). Specifically mixing binding glycan **10** and weakly binding glycan **9** on the same phage yielded a multivalent display that was only weakly binding. The complex interplay of glycan density and glycan structure might explain the inconsistent binding preferences between prior glycan array experiments. These

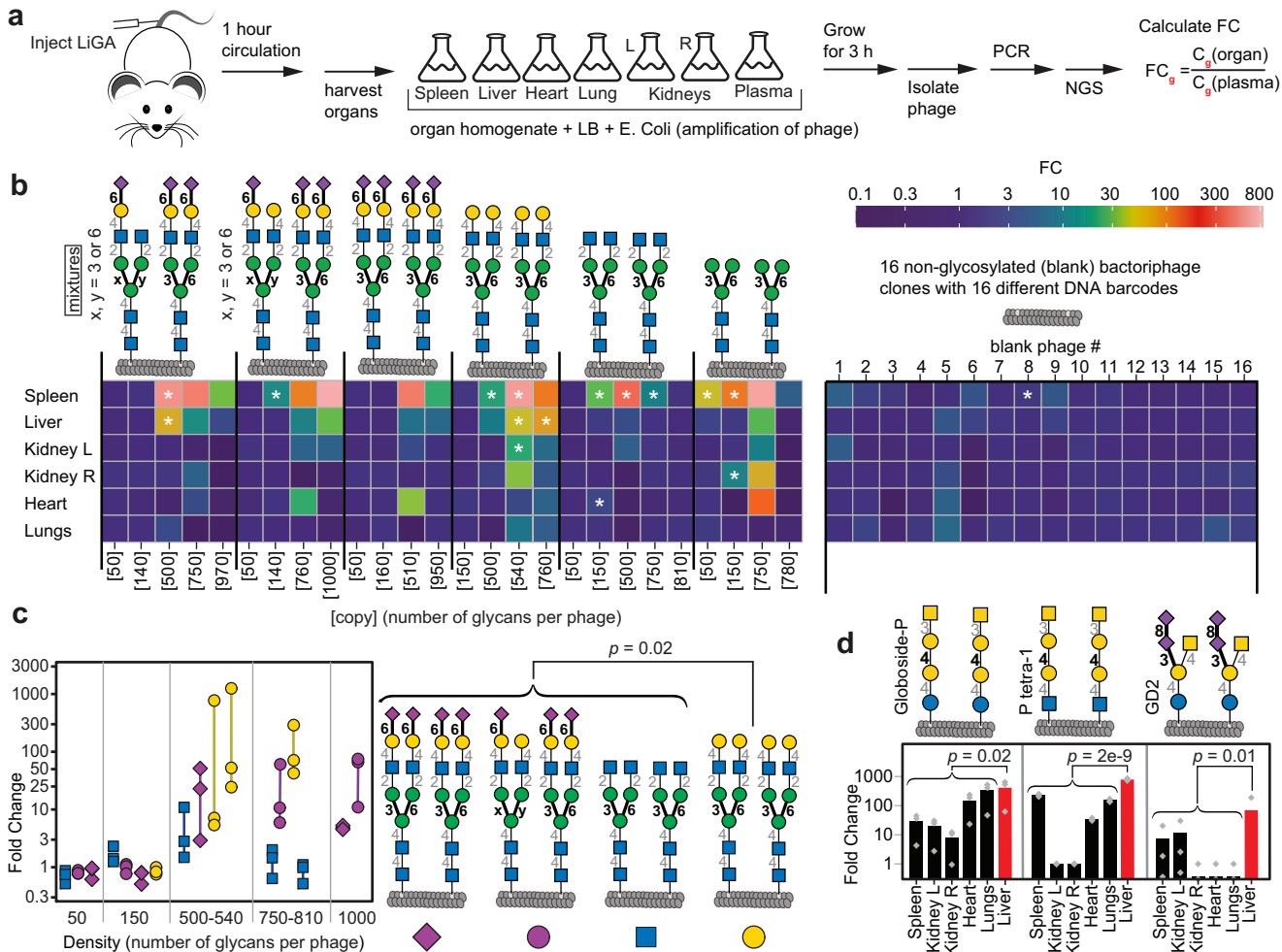

**Fig. 6 | Interaction of LiGA with organs in vivo. a** LiGA was injected (tail vein) into mice (*n* = 3 animals). After 1 h, mice were perfused with saline and euthanized. Plasma and organs were collected; organ homogenates and plasma were incubated with *E. coli* for 3 h to amplify the phage, followed by PCR and NGS of phage DNA. Fold change (FC) between organ and plasma was calculated by edgeR after normalizing by scaling the unmodified phages to unity, and multi-test control by FDR from B-H correction. **b** Heatmap of the enrichment of *N*-glycan-decorated and unmodified (blank) phages in organs. Asterisks designate enrichment with FDR ≤ 0.05. See Supplementary Figs. 50–56 for detailed summaries per organ. **c** Focusing only on the liver data in the heatmap, we summarize enrichment of the four classes of *N*-glycans, *n* = 3 data points represent data from *n* = 3 animals. Fold changes for each glycan in the liver broken down by glycan density. A Mann-Whittney U-test (two-sided) indicates significant (*p* = 0.02) enrichment in the liver of phages with galactose-terminated *N*-glycans when compared to GlcNAc-terminated and completely or partially sialylated *N*-glycans. **d** Phage decorated by three glycans with terminal GalNAc that show significant enrichment (FDR ≤ 0.05) in liver, *p*-values from a Fisher (standard) ANOVA test when compared to other organs. See Supplementary Fig. 51 for summary of other glycans enriched in liver (vs. plasma).

LiGA experiments thus emphasize the importance of testing multiple glycan densities as GBP–glycan binding depends on not only on lectin concentration but also on the spatial arrangement (density) of glycans.

**Binding of LiGA to live animals**

Previously we employed LiGA to measure interactions between glycans and receptors on the surface of B cells in live animals[38]. Here, we injected LiGA into the mouse tail vein (*n* = 3 animals), recovered the organs after 1 h, amplified phages from the organs and plasma and employed NGS of phage populations to determine the structure and density of glycans associated with each organ (Fig. 6 and Supplementary Fig. 50–56). We compared the Fold Change (FC) difference in copy number and its significance (False Discovery Rate, FDR < 0.05) for each glycophage in each organ with respect to the same glycophage recovered from plasma. The injected LiGA contained *N*-glycan glycophages (components of LiGA6×5 in Fig. 5), previously described glycophages that displayed synthetic glycans[38], 10 phage clones in which

the DBCO handle was capped by azidoethanol and 16 unmodified phage clones. The latter 16 + 10 "blank" clones served as important baseline of the in vivo homing experiment and exhibited only a minor fluctuation in FC across all organs (Fig. 6b). The significant (FDR < 0.05) spleen enrichment of diverse *N*-glycans (Fig. 6b) and synthetic glycans (Supplementary Fig. 50) may be attributed to binding to lectins expressed by lymphocytes, macrophages, dendritic cells, and plasma cells residing in spleen. In contrast, few *N*-glycans were enriched in kidneys, heart and lungs; a high FC ratio of paucimannose-conjugated phage was detected in kidneys and heart but its significance could not be inferred; no significant enrichment of any *N*-glycan at any density was observed in the lungs. In liver homing, phage particles that displayed various densities of de-sialylated *N*-glycans with terminal galactose exhibited a significant (*P* = 0.02) accumulation in the liver when compared to *N*-glycans in which galactose was fully or partially capped by sialic acid (Fig. 6b, c). Removal of terminal galactose to expose terminal GlcNAc abrogated liver targeting (Fig. 6b, c). This observation resembled natural clearance of desialylated red

blood cells and platelets by the liver. Specifically, aged, desialylated platelets, are cleared by the hepatic Ashwell Morell Receptor (AMR) complex composed of two asialoglycoprotein receptors (ASGPR) 1 and 2[67,68]. From 12 glycans significantly enriched in liver when compared to plasma (FDR < 0.05), three synthetic glycans—P1 tetra (β-GalNAc-(1→3)-α-Gal-(1→4)-β-Gal-(1→4)-β-GlcNAc), Globoside P (β-GalNAc-(1→3)-α-Gal-(1→4)-β-Gal-(1→4)-β-Glc) and GD2 (β-GalNAc-(1→4)[α-Neu5Ac-(2→8)-α-Neu5Ac-(2→3)]-β-Gal-(1→4)-β-Glc)—enriched in liver significantly more than in any other tested organs ($p < 0.05$). All three contain a terminal β-linked GalNAc residue. We observed that phage clones displaying P1 tetra, Globoside P, GD2 also bound to HepG2 cells, which express ASGPR receptors; the binding was 10–20x higher ($p < 0.001$) when compared to unmodified phage (Supplementary Fig. 57 and Supplementary Data 7). This observation mirrors the well-known delivery of GalNAc-conjugates to ASGPR receptors in the liver used in FDA-approved drugs (Givlaari) and 28 other GalNAc-conjugated oligonucleotides tested in phases I–III of clinical trials[69]. Biodistribution of glycophages—components of LiGA—thus follow a number of well-known biological mechanisms. These results highlight the possibility of using LiGA to identify both the structure and density of glycans necessary for homing of glycoconjugates to a specific organ paving the route to discovery of strategies for delivery of therapeutics and uncovering mechanisms that govern glycan-driven biodistribution in vivo.

## Discussion

The previous LiGA[38] employed 70–90 small synthetic glycans and represented a powerful approach to probe glycan–GBP binding. However, interactions of larger glycans with GBPs cannot always be accurately extrapolated from data obtained with smaller structures[70]. In this paper, we address this gap by expanding the LiGA technology to larger molecules, in particular N-glycans. To increase synthetic efficiency, we developed "on-phage" enzymatic trimming and extension starting from a readily available N-glycan, resulting in a flexible strategy for producing a soluble array in which glycan density can be controlled. A major advantage of this approach is the ability to modify precursors displayed at a defined density leading to products displayed at the same density. Success required developing an analysis strategy where MALDI-TOF MS is used to ensure completion of each enzymatic reaction and to profile and improve reaction conditions. For example, MALDI-TOF MS uncovered that extension and trimming of glycans proceeds more slowly with a higher density glycan precursor (Supplementary Figs. 20, 21). Poor modification of glycans by glycosyltransferases on phages modified by 1000 glycans or 40% occupancy of pVIII proteins (Supplementary Figs. 20, 21) is consistent with the poor accessibility of the same 1000 glycan/phage constructs by most lectins; from twelve tested lectins, nine cease any binding when their glycan epitope are displayed on phage at 1000 copes per phage. Interestingly, four lectins: SiaFind™ α-(2→6)-Specific reagent, SiaFind™ Pan-Specific Lectenz®, CD22, and WGA uniquely increases binding at 1000 copies. Such preference or avoidance of high copy number glycans was confirmed using multi-barcode encoding (Supplementary Figure 49). Grafting of N-glycan X-ray structures to X-ray structure of M13 bacteriophage suggests that packing of glycans at 40% of p8 protein creates dense cylindrical display in which glycan epitopes might not be accessible (Supplementary Fig. 42). We caution that such static grafting should be used with caution as it does capture N-glycan dynamics. This examination of density vs. enzymatic conversion provides insight into enzymatic remodeling of glycans in a complex milieu, here a bacteriophage. To the best of our knowledge, enzymatic remodeling of glycans on bacteriophage surfaces have not been reported. Enzymatic remodeling of glycans in solution has been used by many laboratories[71,72] but applying such reactions to remodeling of immobilized glycans is not trivial. Enzymatic modification of immobile glycans on solid surfaces often show slower enzymatic conversion when compared to analogous conditions in solution[71,72]. Webb, Flitcsh, and co-workers demonstrated that modification of mobile glycans on the surface of liposomes also proceeds significantly slower than modification in solution[58]. The authors elegantly demonstrated that increased density of glycan acceptor on liposome further reduced the rate of modification suggesting that at low density, the decrease may arise from mass-transport limitations of the diffusion of enzyme to the 2D surface of 100–1000 nm liposome whereas steric occlusion plays additional role in liposomes with high density of glycan acceptor[58]. LiGA displays immobile glycans on a 5 × 700 nm cylinder (dimensions of M13 bacteriophage) and the enzymatic modification on phage mirror observations from synthesis of glycans on liposomes. Analogous in-situ MS analysis is difficult for on-glass or on-cell synthesis. On-phage enzymatic modification also provides economic benefits: an order of magnitude decreases in the amount of starting glycan needed to manufacture LiGAs compared to glass-based arrays (see Supplementary Fig. 58).

Using LiGA6×5, we uncovered a previously uncharacterized interplay between N-glycan structure and density in lectin recognition not only in vitro with pure proteins, but also when they are displayed on cells ex vivo and on organs in vivo. For example, the interplay of glycan density and binding of immunomodulatory proteins such as CD22 and DC-SIGN can be used to understand how cellular density of displayed glycans regulate engagement of these lectins. We observe that binding to DC-SIGN(+) cells requires higher density of paucimannose with GlcNAc capping vs. without capping; and similar observations was made in binding of DC-SIGN cells to CHO glycosylation mutant Lec8 that expresses N-glycans[65]. Similar, density-dependent recognition of sialic acids by Siglecs on phage by CD22(+) cells may aid explaining the recognition of low and high densities of sialic acids on cells by Siglec(+) immune cells[73]. It has been long postulated that GBP–glycan binding is dictated not only by glycan structure, but also by their density on the cell surface[22,74,75]. Nevertheless, efforts to address this issue have been limited to specialized solid-phase arrays[19,29] or single glycans displayed on polymers[31] or liposomes[34]. The approach described here, which involves deploying a range of glycans at different densities in a single experiment, provides a more complete picture of glycan–GBP interactions. Such experiments help to explain discrepancies in previous GBP–glycan binding data originating from different arrays, which we postulate arise, in part, from density effects. In this paper we use the term "density" to denote the degree of modification of pVIII proteins by glycans (500 out of 2700 available, 750 out of 2700 available, etc.). While the distances between the pVIII proteins are rather uniform, the distances between the glycans fluctuated around the average, with some local regions where the glycans are closer and further apart than on average (Supplementary Fig. 36). There are only a few multivalent scaffolds in which such fluctuations are augmented: notable exceptions are short rigid DNA molecules that carry glycans on two ends[76], well-packed and defect-free self-assembled monolayers of glycans on gold[77]; bacteriophages in which most capsid protein has been modified by a glycan[20] and bespoke 3D multivalent displays of glycans on folded miniproteins or aptamers[78,79]. However, fluctuations of glycan-to-glycan distances are unavoidable for multivalent scaffolds in general either due to stochastic substitution or flexibility of the scaffold[80]. Such scaffolds in principle mimic lectin–glycan interaction in physiological conditions in the presence of such the local variations in distances between the glycans on the cell surface[24].

LiGA constructs—glycosylated bacteriophages or "glycophages"—like polymers, liposomes, hollow bacteriophage Qβ capsids[20,35] or protein cages[81] have an optimal 'soluble' format suitable for cell-based assays but differ from these more traditional formats, as they can be genetically encoded. Other cell-based glycoarrays[49,52,82] can potentially give rise to similar, DNA-encoded multivalent constructs displayed in a natural milieu. However, precise control of mean density and

composition of a desired glycan structure in cell-based approaches is difficult. LiGA thus represents a heretofore unavailable tool to study GBP–glycan interactions in vitro and in vivo, which combines ease of manufacture and characterization, with the ability to display and encode diverse synthetic and natural glycans at defined density.

## Methods

### Materials and general information

Pronase from *Streptomyces griseus* was purchased from Roche. Neuraminidase from *Clostridium perfringens* and galactosidase from *Aspergillus niger* were purchased from Sigma–Aldrich. β-N-Acetylglucosaminidase S from *Streptococcus pneumoniae* was purchased from New England BioLabs. The glycosyltransferases Pd2,6ST and B4GalT1 were expressed and purified as described in Supplementary information. Concanavalin-A (ConA, #C2010) and Wheat Germ agglutinin (WGA, #L9640) and *Erythrina cristagalli* lectin (ECL, #L5390) were purchased from Sigma–Aldrich. *Galanthus nivalis* lectin (GNL) was purchased from GlycoMatrix (#21510244-1). *Sambucus nigra* agglutinin (SNA-1), *Lens culinaris* agglutinin (LCA) was from EY laboratories, *Pisum sativum* agglutinin (PSA) was from GlycoMatrix, *Ricinus communis* agglutinin I (RCA-I), Carbohydrate-binding modules (diCBM40) were a generous gift from Prof. Lara Mahal (University of Alberta). SiaFind™ α-(2→6)-specific reagent (#SK2601B), SiaFind™ Pan-Specific Lectenz® (#SK0502B), were purchased from Lectenz Bio. Sanger sequencing and deep sequencing was performed at the Molecular Biology Service Unit (University of Alberta) using an Illumina NextSeq500 system. The DC-SIGN(+) and DC-SIGN(-) rat fibroblast cells were a gift from K. Drickamer (Imperial College, London). The CD22(+) and CD22(-) Chinese Hamster Ovary cells were obtained from Macauley and co-workers (University of Alberta). The HepG2 cells were a generous gift from Ravin Narain (University of Alberta). All DNA primers were ordered from Integrated DNA Technologies. Biochemical reagents were purchased from Thermo Fisher Scientific unless noted otherwise. HEPES buffer contains 20 mM HEPES, 150 mM NaCl, 2 mM $CaCl_2$, pH 7.4. PBS buffer contains 137 mM NaCl, 10 mM $Na_2HPO_4$, 2.7 mM KCl, pH 7.4. Solutions used for phage work were sterilized by filtration through 0.22 μm filters.

### SDB clone isolation and amplification

The M13-SDB-SVEKY library described in a previous report[38] was used to isolate individual phage clones with built-in silent double barcodes, this is a short, readily PCR amplified, region of the phage genome that varies between clones subject to the restriction that all variants code for the same amino acid sequence.

### Analysis of glycosylation of phage samples by MALDI-TOF MS

MALDI-TOF MS spectra were recorded on ABI Sciex Voyager Elite MALDI MS Voyager instrument control panel version 5.0 equipped with a MALDI-TOF pulsed nitrogen laser (337 nm) (3 ns pulse up to 300 μJ/pulse) operating in Full Scan MS in positive ionization mode. Nanodrop™ (Thermo Fisher) was used to measure the absorbance of protein and DNA solutions. A sinapinic acid matrix was formed by deposition of two layers. Layer 1 was prepared from a solution of sinapinic acid (Sigma, #D7927, 10 mg/mL) in acetone–methanol (4:1). Layer 2 was prepared from a solution of sinapinic acid (10 mg/mL) in acetonitrile–water (1:1) with 0.1% TFA. In a typical sample preparation, 2 μL of phage solution in PBS was combined with 4 μL of Layer 2, then 0.7 μL of Layer 1 was deposited. Once completely dry, 1.5 μL of the Layer 2+phage mixture was deposited on top. Spots were washed with 0.1% TFA in water (10 μL) to remove salts from the PBS. To estimate the ratio of modified to unmodified pVIII, we implemented an automated pipeline for processing of raw MALDI *.txt files to images and integration data. This task was performed by plotMALDI.m MatLab script. The data of MALDI plots used in this work is available in Supplementary Data 1/All MALDI Data/ folder.

### Chemical modification of phage clones with glycans to build LiGA components

A solution of SDB phage clone ($10^{12}$–$10^{13}$ PFU/mL in PBS) was combined with DCBO–NHS (20 mM in DMF) to afford a 0.2–2.0 mM concentration of DCBO-NHS in the reaction mixture, which typically yields 5–50% of pVIII modification after 45 min of incubation. After conjugation of DBCO-NHS, each clone was individually purified on a Zeba™ Spin Desalting column (7 K MWCO, 0.5 mL, Thermo Fisher, #89882) following the manufacturer instructions. Solutions of azido-glycans (10 mM stock in nuclease free $H_2O$) were added to the filtrates to afford a 2 mM concentration of the azido-glycan and the solutions were incubated overnight at 4 °C. All chemical reactions were verified and quantified by MALDI-TOF MS as described in previous sections. If reactions were incomplete and a residual pVIII-DBCO peak was detected by MALDI-TOF MS, we added an additional amount of azido-glycan and extended the incubation time. If reactions were complete, the conjugates were purified on a Zeba™ Spin Desalting column and stored at 4 °C or supplemented with glycerol and stored as a 50% glycerol stock at −20 °C. All glycans used in this work to build LiGA components are available in Supplementary Data 5.

### Procedures for phage surface glycosidase treatment

Cleavage of sialic acid on phage-displayed N-glycans by neuraminidase: after modification of phage clones with **7** to afford the conjugated phage (60 μL, ~$10^{12}$–$10^{13}$ PFU/mL in PBS, pH 7.4), 5% PEG-8000, 0.5 M NaCl (12 μL) was added and the solution was kept for 1.5 h at 0 °C. The solution was then centrifuged at 21000 × g for 10 min at rt. The supernatant was removed, and the pellet was resuspended in sodium acetate buffer (40 μL, 50 mM, pH 5.5 containing 5 mM $CaCl_2$) followed by the addition of *Clostridium perfringens* neuraminidase (1 μL, 0.02 units). The reaction was incubated at 37 °C and monitored by MALDI-TOF MS. The reaction was complete in 1 h to afford galactose-terminating *N*-glycans on the phage surface. Procedures for other on-phage glycosidases treatments were described in Supplementary information.

### Procedures for phage surface glycosylation

Installation of α-(2→3)-linked sialic acid using Pm2,3ST: To a solution of phage modified with terminal galactose N-glycan 10 (50 μL, ~$10^{12}$ PFU/mL in PBS, pH 7.4), 5% PEG-8000, 0.5 M NaCl (15 μL) was added and the solution was kept for 1.5 h at 0 °C. After 1.5 h, the solution was centrifuged at 21000 × g for 10 min at rt and the supernatant was decanted. The pellet was resuspended in Tris-HCl buffer (20 μL, 50 mM with 10 mM $MnCl_2$, pH 7.5) containing CMP-Neu5Ac (100 μg), Shrimp Alkaline Phosphatase (0.5 μL) and Pm2,3ST (2 μL, 6.4 mg/mL). The reaction mixture was incubated at 37 °C and monitored by MALDI-TOF MS. The reaction was complete in 1 h and was validated by MALDI-TOF MS (Supplementary Fig. 17). Procedures for other on-phage glycotransferases treatments were described in Supplementary method.

### Preparation of LiGA from glycosylated clones

In a typical protocol, a LiGA was prepared by mixing $10^8$ PFU of desired glycan-phage conjugates in a single tube. The mixture was characterized by titering and N×$10^6$ PFU (N = glycan-phage conjugates) was used for a typical lectin or cell-binding experiment. Each unique LiGA mixture was assigned a two-letter identifier (e.g., "SC") and a "dictionary" prepared (e.g., SC.xlsx), a table that describes the correspondence between the DNA barcode sequences and the glycans in the LiGA mixture (Supplementary Data 4). These dictionaries were subsequently used to translate from nucleotide sequences in the deep-sequencing files to the corresponding glycans (including density). Examples of dictionaries used are available as part of the Supplementary Data 4. In characterizing the LiGA mixture by deep sequencing, we noted that the mixing of the solutions matched by the titer of phage stock did not afford uniform distribution of barcodes after

sequencing. The composition of each naïve library or a naïve library binding to a control target, thus, was used as normalization factor in each experiment. Dictionaries for LiGA mixtures relevant to this manuscript are available in Supplementary Data 4/LiGA Dictionaries/ folder. Naïve compositions of these mixtures and references to the deep-sequencing data are available on https://48hd.cloud/. with dataset name listed in Supplementary Table 1. To access the data, enter the dataset name at https://48hd.cloud home page search bar. Locations of "silent barcode" regions SB1 and SB2 in the M13 genome is illustrated in a previously published work[38].

### Binding of LiGA to lectins immobilized on plate

Lyophilized Concanavalin A (Sigma-Aldrich, #C2272) was dissolved in HEPES at final concentration of 1 mg/mL. The solution was then diluted with HEPES to afford a final concentration of 20 μg/mL and 50 μL was added to each well of a 96-well plate (Corning®, #CLS3369). The plate was covered with sealing tape (Thermo Scientific™, #15036) and incubated overnight at 4 °C. The following day, the wells were washed 3x by adding washing buffer (200 μL, 0.1% Tween-20 in HEPES) in the wells and discarding the solution by inverting the plate on top of a paper towel. Thereafter, blocking solution (100 μL, 20 μg/uL BSA in HEPES) was added to the wells and incubated for 1 h at rt. The solution was discarded by inverting the plate, the wells were then washed three times with washing buffer. After the incubation with blocking solution and triple washing, 50 μL of LiGA6×5 ($8×10^8$ PFU/mL in HEPES) was added to the wells. The solution was incubated for 1 h at rt and discarded by inverting the plate. The wells were washed 2x with washing buffer and 1x with HEPES (200 μL). To elute bound phage, 50 μL of HCl (pH 2.0) was added to the well, incubated for 9 min at rt, and the content of each well was transferred to an Eppendorf tube containing 25 μL of 5× Phusion HF buffer (NEB, #M0530S). The neutralized solution was used for titer and as DNA template for PCR and Illumina sequencing. Binding of LiGA to other lectins was performed in similar fashion.

### Binding of LiGA to CHO cells expressing CD22

Production and maintenance of CHO cells expressing CD22 were described previously[38]. The binding procedure is adapted from a previous publication[38]. Confluent CHO-CD22(+) and CHO-wt cells were detached from culture flask using PBS containing EDTA (5 mM) and centrifuged for 5 min at 300 × g. The supernatant was decanted and the pellet was washed twice by resuspending it in HEPES (5 mL) and centrifuged for 5 min at 300 × g. After final washing, the cell pellet was resuspended in incubation buffer (1% BSA in HEPES buffer) at $2×10^6$ cells/mL. The cells were aliquoted (500 μL) to a FACS tube (Corning, #352054), which afforded 1 million cells per FACS tube. Thereafter, LiGA was added to each FACS tube at $10^8$ PFU, which afforded approximately $10^6$ PFU of each clone in the incubation solution. The solution was incubated for 1 h at 4 °C. After incubation, the cells were gently vortexed (Speed 1, Fisher Vortex Genie 2™, #12-812) and wash buffer (3 mL, 0.1% BSA in HEPES buffer) was added to each FACS tube using a small squirt bottle. The solution was centrifuged at 218 × g for 5 min at 4 °C in a swinging bucket rotor. The supernatant was decanted by inverting the FACS tubes and blotting on a Kimwipe. Two additional washes were performed: during each wash, the tube was filled with 3 mL of wash buffer, centrifuged and inverted to discard the supernatant in same manner as described above. After the last wash, the pellet was resuspended in 1 mL of HEPES buffer, transferred to a microcentrifuge tube and centrifuged for 5 min at 218 × g at 4 °C in a swinging bucket rotor. The supernatant was discarded by pipetting and the pellet was resuspended in nuclease-free water (30 μL). An aliquot of this solution (2 μL) was sampled and combined with PBS (500 μL, pH 7.4) for tittering. The remaining solution was incubated at 90 °C for 15 min, centrifuged at 21,000 × g for 10 min, and the supernatant (25 μL) was used as template for PCR reaction as described in PCR protocol section.

### Binding of LiGA to fibroblasts cells expressing DC-SIGN

Production and maintenance of rat fibroblast cells expressing DC-SIGN was described previously[38]. The binding procedure is adapted from a previous publication[38]. The Rat-6 fibroblast DC-SIGN(+) and DC-SIGN(−) fibroblast cells were detached from a culture flask using TrypLE (ThermoFisher, # 12605036) and resuspended in incubation buffer (20 mM HEPES, 150 mM NaCl, 2 mM $CaCl_2$, pH 7.4, 1% BSA,) to produce $2×10^6$ cells/mL suspension. The cells were aliquoted (500 μL) to a FACS tube (Corning, #352054), which afforded 1 million cells per FACS tube. Binding of LiGA to these cells was performed using the steps identical to binding of LiGA to CHO cells expressing CD22.

### PCR protocol and Illumina sequencing

The 15 μL of DNA template solution in Nuclease free water was amplified in total volume of 50 μL with 1x Phusion® buffer, 50 μM of each dNTPs, 500 μM $MgCl_2$, 1 μM forward barcoded primer (IDT), 1 μM reverse barcoded primer (IDT) and one unit of Phusion® High-Fidelity DNA Polymerase (NEB, #M0530S). Sequence of all primers used in this work are provided in Supplementary Data 6. The volumes and PCR cycling parameters are further detailed in the in Supplementary methods. The PCR products were quantified by 2% agarose gel in Tris-Borate-EDTA buffer. PCR products that contain different indexing barcodes were pooled allowing 10 ng of each product in the mixture. The mixture was purified by eGel, quantified by Qubit (Thermo Fisher) and sequenced using the Illumina NextSeq paired-end 500/550 High Output Kit v2.5 (2 × 75 Cycles). Data was automatically uploaded to BaseSpace™ Sequence Hub. Processing of the data is described in section "Processing of Illumina data".

### Panning of LiGA in mice

All procedures and experiments involving animals were carried out using a protocol approved by the Health Sciences Laboratory Animal Services (HSLAS) at the University of Alberta. The protocol was approved as per the Canadian Council on Animal Care (CCAC) guidelines. All mice were maintained in pathogen-free conditions at the University of Alberta breeding facility. The strain of each mouse is C57BL (no substrain). Three female mice were used in this experiment and their age is around 10–12 weeks old. Each mice were injected with LiGA (0.2 mL, $1×10^{11}$ PFU/mL in PBS). One-hour post-injection mice were euthanized with $CO_2$ and blood (0.5 mL) was drawn and stored on ice. Internal organs (heart, liver, kidneys, lungs, and spleen) were collected and stored in cold DMEM (Thermo Fisher). Tissues were homogenized by grinding between 75 mm frosted microscope slides. Homogenized tissues of each organ were transferred into 25 mL LB and supplemented with a 0.5 mL of log phase E. coli K12 ER2738. After incubation at 37 °C for 3 h, the amplified phage in the culture were isolated by centrifugation at 4500 × g for 10 min. The supernatant was incubated with 5% PEG-8000, 0.5 M NaCl for 8 h at 4 °C, followed by 30 min centrifugation at 13,000 × g. The phage pellet was resuspended in 1 mL PBS-Glycerol 50% and solution of phages (2 μL) was PCR amplified using barcoded sequencing primers using the protocol described above in "PCR Protocol" and analyzed by Illumina sequencing.

### General data processing methods

General data processing methods were reported in a previous paper[38]. Data analysis was performed in Python, Matlab or R. Comparison and testing differences for significance in LiGA data was performed essentially as differential enrichment analyses of phage displayed libraries described in our previous reports[83,84] and based on differential expression (DE) analysis implemented in edgeR[83,84]. The DE-analysis has three major factors were considered: (i) the abundance of each phage/glycan is estimated from the observed counts using a negative binomial model; (ii) the abundances between two sets of samples can be tested for significant differences, in order to deal with the many

such tests, Benjamini–Hochberg (BH) adjustment is used to control the false discovery rate (FDR) at α = 0.05[84]; (iii) normalization of data across multiple samples assumes that a group of the phages (blank/unlabeled phages for in vivo data and azidoethanol-labeled phages for in vitro) is invariant between the test and control data. Alternatively for the CD22 cell data where there were few azidoethanol reads, normalization was instead by Trimmed Mean of M-values (TMM)[84]. Core processing scripts are available as part of the Supplementary information or on GitHub. To assess the significance of a glycan binding in a specific experiment the differential enrichment of the levels of the DNA barcode associated with that glycan in "test" sets of DNA read was compared to the levels of the same barcode in "control" sets. For example, in cell-based experiments, the "test" dataset was from an association of a LiGA mixture with receptor (+) cell line, whereas the control dataset was from association of the same LiGA mixture with the isogenic cell line that contains no target receptor. In binding to lectins, the "control" dataset was from association of the same LiGA mixture with blank carriers (BSA-coated wells). Prior to DE-analysis, "test" and "control" data sets were retrieved from the http://48hd.cloud/ server as tables of DNA sequences, and raw sequencing counts (Supplementary Table. 1). Test and control data sets of Figs. 5 and 6 are also available in Supplementary Data 2 and 3. DNA reads that could not be mapped to any entries in the LiGA dictionary were discarded.

### Processing of Illumina data

The processing of FASTQ files downloaded from BaseSpace™ Sequence Hub to reads and frequencies of these glycans was performed as previously described[85–87]. A LiGA-specific lookup table ("LiGA dictionary") was used to convert the identified SDB to glycans and display density. Abbreviated names of glycans are based on those recommended by the Consortium for Functional Glycomics: http://www.functionalglycomics.org/static/consortium/resources/resourcecored2.shtml.

Translated files with raw DNA reads, raw counts, and mapped glycans were uploaded to a server, https://48hd.cloud/. All LiGA sequencing data is publicly available at this location. Each experimental dataset has a unique alphanumeric name (e.g., 20220314-87SCwgacBI-CT, see Supplementary Table 1).

### Native MS methods to measure binding affinity of *N*-glycans

CaR-ESI-MS screening was performed in negative ion mode using a Q Exactive Ultra-High Mass Range Orbitrap mass spectrometer and the affinity measurements were carried out in positive ion mode on a Q Exactive Orbitrap mass spectrometer (Thermo Fisher Scientific, Bremen, Germany). Both instruments were equipped with a nanoflow ESI (nanoESI) device. NanoESI tips were produced from borosilicate glass capillaries (1.0 mm outside diameter (o.d.), 0.78 mm inner diameter (i.d.), 10 cm length or 1.2 mm o.d., 0.69 mm i.d., 10 cm length) using a P-1000 micropipette puller or a P-97 micropipette puller (Sutter Instruments, CA). Raw data were processed using the Thermo Xcalibur 4.2 software. Time-resolved mass spectra were averaged over 1 min intervals (https://github.com/pkitov/CUPRA-SWARM)[88]. For glycoproteins, the abundance was determined using charge state-normalized peak areas that were determined by fitting the ion signal with Gaussian functions using the IgorPro Multipeak Fitting 2 tool (WaveMetrics Inc., Lake Oswego, OR, USA).

### Direct ESI-MS affinity measurements (for CD22, ConA, ECL, WGA)

The affinities were measured by ESI-MS in positive ion mode using a Q Exactive Orbitrap mass spectrometer (Thermo Fisher Scientific, Bremen, Germany). The measurements were performed using a sub-micron nanoESI emitters[89] and a spray voltage of 0.4 kV. The inlet capillary temperature was 180 °C and the S-lens RF level was 100; an

automatic gain control target of $5 \times 10^5$ and maximum injection time of 200 ms were used. The resolving power was 17,500. For a given GBP-glycan system, the dissociation constant ($K_d$) was determined by fitting Eq. 1 to a plot of fraction bound protein ($R/(R+1)$) versus initial ligand concentration $[L]_0$:

$$\frac{R}{R+1} = \frac{[P]_0 + [L]_0 + K_d - \sqrt{(K_d - [L]_0 + [P]_0)^2 + 4K_d[L]_0}}{2[P]_0} \quad (1)$$

where $[P]_0$ and $[L]_0$ are the initial protein and ligand concentrations, respectively, and $R$ is the total ion abundance ($Ab$) ratio ($R$) of the ligand bound (PL) to free protein (P), which is taken to be equal to the solution concentration ratio (Eq. 2):

$$R = \frac{[PL]}{[P]} = \frac{Ab[PL]}{Ab(P)} \quad (2)$$

### Concentration-independent catch-and-release native mass spectrometry affinity measurements (for SNA and RCA)

For SNA and RCA-I, affinities were measured using the Concentration Independent Catch-and-Released native mass spectrometry (COIN-CaR-nMS) method using a Q Exactive Ultra-High Mass Range Orbitrap The method exploits the Catch-and-Released (CaR)-ESI-MS technique in combination with slow mixing of two solutions with different glycan concentrations and mathematic model to obtain $K_d$[90–92]. Measurements were performed using the Q Exactive Ultra-High Mass Range Orbitrap mass spectrometer. The inlet capillary temperature was 180 °C and the S-lens RF level was 100; an automatic gain control target of $5 \times 10^5$ and maximum injection time of 200 ms were used. The resolving power was 25000. Collision energy of 80 V was used.

### Chemical synthesis

All reagents were purchased from commercial sources and used without further purification. Oven-dried glassware was used for all reactions. Reaction solvents were dried by passage through columns of alumina and copper under nitrogen. All reactions, unless stated otherwise, were carried out at rt under positive pressure of argon. Organic solutions were concentrated under vacuum below 40 °C using a rotary evaporator. Reaction progress was monitored by TLC on Silica Gel 60 $F_{254}$ (0.25 mm, E. Merck). Visualization of the TLC spots was done either under UV light or charring TLC plates with acidified *p*-anisaldehyde solution in ethanol or phosphomolybdic acid stain. Column chromatography was performed using silica gel 40–60 μm. [1]H NMR spectra were recorded using 500 MHz or 400 MHz instruments, and the data are reported as if they were first order. [13]C NMR (APT) spectra were recorded at 125 MHz. Reactions performed in solution were confirmed by LC-MS. The detailed information of LC-MS method is described in the Supplementary information. Details of the synthetic procedures are described in the Supplementary information.

### Reporting summary

Further information on research design is available in the Nature Portfolio Reporting Summary linked to this article.

## Data availability

All raw deep-sequencing data is publicly available on https://48hd.cloud with dataset name listed in Supplementary Table 1 and 2. To access the data, enter the dataset name at https://48hd.cloud home page search bar. All sequence count data and its analysis are provided as a Supplementary Data 2 and 3. CFG data used in the Supplementary information can be downloaded from http://www.functionalglycomics.org. MALDI-TOF data is provided as part of Supplementary Data 1.

## Code availability

MatLab, Python and R scripts used in this work to perform data analysis are available on GitHub and archived in Zenodo at https://doi.org/10.5281/zenodo.8188084[93]. Here is the URL to access the scripts: https://doi.org/10.5281/zenodo.8188084.

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

## Acknowledgements

We thank the staff at the University of Alberta mass spectrometry facility (Chemistry Department) for help with MALDI analysis and Sophie Dang at the molecular biology service unit for assistance with Illumina sequencing. We thank Lara Mahal (University of Alberta), for provision of critical reagents. We thank Sussex Research Laboratories Inc. for the generously gifting Tri-GalNAc compound. The authors acknowledge funding from NSERC (RGPIN-2018-04365 to T.L.L., RGPIN-2018-03815 to M.S.M., and RGPIN-2016-402511 to R.D.) and NSERC Accelerator Supplement (to R.D.), CIHR (#180445 to R.D.), GlycoNet (SD-1 to T.L.L., TP-22 to R.D.), Alberta Innovates Strategic Research Project to R.D., and NIH projects (AI118842 to M.S.M., R35GM139643 to P.W., and OD024964 to L.Y.). Many compounds were prepared by the Consortium for Functional Glycomics supported by NIH GM061126. Infrastructure support was provided by CFI New Leader Opportunity (to R.D. and M.M.). E.S.H. and E.A.V. acknowledges summer research fellowship from GlycoNet.

## Author contributions

C.-L.L. performed the synthesis of N-glycans, enzymatic reaction of N-glycans on phage, model study of enzymatic reactions on phage, expression of two glycosyltransferases, density modification of N-glycans on phage, MALDI analysis of reactions and all protein (lectin)-binding experiments. M.S. performed the cell surface lectin binding experiments, modifications of phage by O-glycans. E.J.C. wrote the R scripts and performed statistical analysis of protein-binding and animal experiments. C.-L.L., E.S.H., and S.S. performed animal experiments. E.S.H., and E.A.V. performed amplification of phage clones. E.J.C. performed custom analysis of public CFG data. C.W and P.W prepared two glycosyltransferases. L.Y. provided reagents from Lectenz Bio. D.T.B and J.S.K performed MS binding studies. R.D., and C.-L.L. wrote the manuscript, R.D., T.L.L., and M.S.M. edited the final manuscript and contributed intellectual and strategic input. R.D., and T.L.L. supervised the project. All authors approved the final manuscript.

## Competing interests

R.D. is the C.E.O. and a shareholder of 48Hour Discovery Inc., the company that licensed the patent (WO2018141058A1) describing LiGA technology. The remaining authors declare no competing interests.
