## [Peer Review File · Nature Communications]

REVIEWER COMMENTS

Reviewer #1 (Remarks to the Author):

This manuscript follows on a previous paper by the group (Ref 39) that developed an ingenious phage display platform technology for liquid display of glycan structures. Whereas the previous paper was the first proof-of principle using small synthetic glycans but also demonstrating application in animal models, here more complex N-glycans are displayed and investigated.

The synthetic approach and analysis are very impressive and are based on state-of-the-art methodologies used for N-glycans. Given the difficulty in synthesising the N-glycan core, using the egg glycan which is available in large quantities as starting material makes sense. The pleasingly, the analytics that can be applied here allow for optimisation of the remodelling methods and the data look very convincing. The issues in dealing with analysis of sialosides is commonly encountered and perhaps the authors should discuss the efforts by the Wuhrer group and others to stabilise sialosides for MALDI analysis.

Overall, the paper is very impressive and the data look good.

My only concern is the presentation to a more generic readership - as a glyco scientist I have really enjoyed reading it and finding out about the details of lectin binding and glycoenzyme activity. I just wonder if the more generic messages could have been discussed in more detail.

For example, I was fascinated by the report that glycoenzyme activity relates to substrate density - the authors state that this is a first (in this context they might be interested in the work by Simon Webb (eg doi.org/10.1039/C7CC09148F). The data supporting these finding appear to be buried in the Supplementary Information, which is odd since it reads in the discussion as one of the major findings (paragraph 1 page 19). It would be good if this statement was supported by a more detailed presentation and analysis of results in the main manuscript.

Similarly, a discussion summarising the lectin binding results in a broader context- main messages in terms of biological significance - might be useful to a general reader.

Reviewer #2 (Remarks to the Author):

This study addresses two interesting issues in the field of glycobiology, namely how are glycans recognized and what proteins bind to them. To this end, the authors illustrate here a phage-display approach in which glycan presentation is created with what is termed different densities, and an approach is used to tag such glycans on phage in what is termed 'Genetically-Encoded Multivalent Liquid N-glycan Arrays'.

While the title is a bit misleading, as discussed below, the principle explored here to generate 'glycoconjugates' on phage (restricted to a few Asn-linked or N-glycans) at different densities of conjugation and test their binding in vitro to several lectins or carbohydrate-binding proteins, and also inject them into mice and observe the locations of homing for these derivatized phage.

The work is interesting to a degree, although somewhat preceded by rather similar approaches by this group, and limited to only a few observed glycans or lectins. In essence the study addresses the issue of glycan multivalency and how it affects binding of lectins. The results are not particularly unpredicted, though there are some unusual findings, yet also somewhat inconsistent results within the study. A prior study recently published in Nature Chem. Biol., entitled "Genetically encoded multivalent liquid glycan array displayed on M13 bacteriophage" presents the general approach. The current manuscript does not appreciably extend the observations in novel ways beyond that work. So a major question that this reviewer considered, is that no doubt there is a lot of work to produce the results enclosed herein, but are the results of these studies mainly predictable based on prior studies in the field? On the whole, most of the conclusions of the study are not particularly novel or unpredictable.

1. There are several conceptual issues that trouble this reviewer. While accepting the fact that this work is both interesting and obviously a lot of effort went into it, and what was done here, was done extremely well, the study falls short in impact and mechanism.

2. To start with, the title of the paper is a bit misleading, in terms of indicating that they have constructed an "N-glycan library" and an "Array" and that it is "Genetically-Encoded". The prior paper cited above generated a liquid glycan array (LiGA) platform, and used a total of 140 glycan-modified phage, while the current paper under review here only used "...a library of six N-glycans". The argument could be made that this limited number is studied in a different way, i.e. density, from the prior studies, but again, with so few glycans and lack of controls (isomeric glycans as mentioned below), the conclusions are not rigorously supported.

3. The authors only study a total of 6 glycans, hardly a 'library' in the literal sense, and these 6 glycans lack diversity, which is hardly impressive. First, they are all N-linked, and no 'isomers' of glycans are used, as in the obvious alpha2-6 sialic acid versus alpha2-3 sialic acid modifications. Differential recognition of specific glycan structures that are isomeric in structure would be more appropriate to gauge any level of specificity. More glycans would also be helpful to make an argument about a library.

4. Much of the work describes the enzymatic modification of glycans on the phage, and while somewhat interesting, the results are rather predictable. For example, galactosyltransferase is shown to modify a terminal GlcNAc-containing N-glycan (Fig. 5), partly and then to completion. Such reactions as shown in this manuscript have been used by many laboratories for many years, so the preparation of the modified glycans and most of the information about this lacks novelty. Similar predictability applies to removal of glycans, as the observation that neuraminidase can completely remove sialic acid, which would be expected, of course (Fig. 2).

5. The concept of density control is not strictly accurate to this reviewer, as the illustrations indicate, the density is not uniform, but randomly varied and a lot of different densities are present as a result of the extent of modification. Such site density effects have been explored with neoglycoproteins in the past made on bovine serum albumin with random degrees of modification, and similar arguments or qualifications can be made for those. The better term might be degree of modification (500, 750 or 1000 glycans) (see Fig. 9).

6. The paper seems more of a methods development, but not especially novel in that case either, as the prior published paper basically provided that technology. The simple array itself is only genetically-encoded in that phage DNA is encoded, not the glycan, and the entire technology of employing phage virions with DNA barcodes in phage genome was described.

7. A lot of the paper used prior methods and lacks originality. The method of preparing the Multivalent Liquid N-glycan Array is not novel, as it is created by the same approach as used previously using alkyl-azido linkers to dibenzocyclooctyne (DBCO)-modified M13 phage. The method of remodeling of the egg-derived sialoglycopeptide is by conventional techniques.

8. In regard to what this reviewer considers the two major discoveries of this work, is that at extremely high conjugation levels some lectins did not bind the phage. The explanation for this is not really provided, other than “presumably due to steric occlusion of tightly packed epitopes”, which is not really an explanation, but a restating of the observation.

9. The second major observation arises from the experiments in which LiGA are injected into the mouse tail vein, and phage levels are determined in recovered plasma and organs after 1 hour. The authors found that many modified phage accumulated in spleen, which is attributed, but not shown, to be due to the abundance of endogenous lectins in lymphocytes, and other accumulated in liver, presumably due to the liver asialoglycoproteins receptor (ASGPR), but this is also not proven.

10. display platform with glycans has already been published, as in the cited recently developed liquid glycan arrays (LiGAs) (reference 39).

11. The identification of glycans recognized in this approach using in vitro approaches was limited to 10 lectins. No studies were performed on the endogenous lectins in mice that might be responsible for homing of conjugates.

12. The artificial approach of presenting clustered glycans on a phage-display approach may not represent any biological situation, and indeed the authors do not identify any specific or known interactions that represent displays of density versus binding that they encounter with this phage display method.

13. The authors make a misleading statement that “The affinity between monovalent glycans and a GBP binding site is typically low”. This type of statement is often made, but applies to non-biological or small oligosaccharides, as binding of GBPs to natural glycan ligands are typically relatively high affinity in the nm to low micromolar range. Of course, by clustering poor ligands, one can increase avidity, but whether this is truly biological representative is not well documented. It seems to be rather a belief. In any case, the authors do not actually describe affinity constants of binding, so this aspect of the work is lacking.

Reviewer #3 (Remarks to the Author):

Glycan-protein interactions play important roles in many biological processes, but methods to study them have key limitations. One of the most common and powerful approaches is glycan array technology. Standard arrays typically contain many different glycans immobilized on a glass slide. Some alternatives include bead-based arrays and cell-based arrays. The Derda group has pioneered a new format – glycan libraries on genetically encoded phage particles. It’s an excited new approach that allows analysis of samples that are not compatible with solid supports like glass. For example, a phage library can be injected into an animal to evaluate localization of glycans to many different parts of the body in parallel.

In this paper, the authors describe an advance to their phage array approach. A key challenge is getting glycans for the array. In this study, they address this limitation by developing methods to attach one or more glycans to the phage and then diversify that library on phage by enzymatically extending and/or trimming the glycans. It’s an efficient, versatile strategy. One really nice aspect of their study is their ability to follow enzymatic reactions by mass spectrometry. It can be quite difficult to track reaction outcomes/progress with large particles, but it works really well for their system. The ability to monitor

progress allowed them to optimize the reactions in ways that are normally quite difficult. As a result, they were able to develop conditions to get essentially complete reactions. This is impressive because enzymatic glycan extension and trimming can often be incomplete, leaving mixtures. As an added layer of information, they also investigate effects of glycan density on the phage. In the end, they built a library of 6 N-linked glycans at 5 different densities for 30 combinations. They then used it to profile a variety of lectins. Lastly, they injected them into mice to track organ localization – something that is impossible with other standard array formats.

Overall, it is a pretty interesting study. After clarifying a few things (see below), it will be a nice addition to Nature Comm.

Revisions

I may have missed it, but how did they authors account for multiple comparisons when doing their statistical analyses? Perhaps add some info in the supporting information.

Figure 6. the legend lists b twice. For the second listing, I don't really understand the Mann Whitney part – what part of the figure are they referring to and which things are significantly different? Should there be asterisks and brackets somewhere? In part d, having 6 bar graphs and 1 p value is a little confusing. what are the specific differences that are significant?

The phage array SAR for SNA looks pretty surprising. Their data suggest that a terminal galactose on one arm plays a key role in recognition. Its absence or extension significantly hampers binding. If the authors think this is real, I would suggest mentioning it in the main text with some discussion, ie. is this known or new, and why would it have a big effect. If they think it is an artifact or just variability of the measurement, I would mention that in the supporting information.

Some of the ConA data is surprising too. The density effect for ConA with Man3 is a lot larger than I would have expected. Also, the lack of binding to Man3GlcNAc2 is surprising given the Cummings and CFG data. Any explanation for these effects? a little more discussion in the main paper could be helpful here. Are there key differences in the platforms leading to these differences in binding profiles?

The GNA binding profiles are also pretty unusual. I find it very surprising the binding would be observed for Man3 at 150/phage and 750/phage but not 500/phage. Any explanation? Is this an experimental

issue/artifact? Maybe some sort of barcode issue or processing issue? Perhaps some independent verification- can you do phage ELISA with these to confirm the results?

The WGA binding to Man3 is pretty surprising too. It seems unusual to have good binding at a density of 750/phage but not 500 or 780. I'm also surprised that it is one of the best binders at 5 and 1ug/mL. Is that in line with prior published data for WGA?

In the liver, spleen and kidney data, the differences between 6'SL at 650 vs 760/phage are pretty surprising. Are these the same glycan at slightly different densities, or am I reading this wrong? Assuming they are the same glycan, why would a density difference of 650 vs 760 have such a large effect? Is it possible that the differences are related to some other aspect of the experiment – like processing side?

REVIEWER COMMENTS

Reviewer #1 (Remarks to the Author):

RI.1. *This manuscript follows on a previous paper by the group (Ref 39) that developed an ingenious phage display platform technology for liquid display of glycan structures. Whereas the previous paper was the first proof-of principle using small synthetic glycans but also demonstrating application in animal models, here more complex N-glycans are displayed and investigated.*

The synthetic approach and analysis are very impressive and are based on state-of-the-art methodologies used for N-glycans. Given the difficulty in synthesising the N-glycan core, using the egg glycan which is available in large quantities as starting material makes sense. The pleasingly, the analytics that can be applied here allow for optimisation of the remodelling methods and the data look very convincing. The issues in dealing with analysis of sialosides is commonly encountered and perhaps the authors should discuss the efforts by the Wuhrer group and others to stabilise sialosides for MALDI analysis.

Overall, the paper is very impressive and the data look good.

We thank reviewer for the very positive feedback and have addressed each of the specific comments as described below.

RI.2. *My only concern is the presentation to a more generic readership - as a glyco scientist I have really enjoyed reading it and finding out about the details of lectin binding and glycoenzyme activity. I just wonder if the more generic messages could have been discussed in more detail. For example, I was fascinated by the report that glycoenzyme activity relates to substrate density - the authors state that this is a first (in this context they might be interested in the work by Simon Webb (eg doi.org/10.1039/C7CC09148F). The data supporting these finding appear to be buried in the Supplementary Information, which is odd since it reads in the discussion as one of the major findings (paragraph 1 page 19). It would be good if this statement was supported by a more detailed presentation and analysis of results in the main manuscript.*

We made the following changes:

1. We added clarification to Supplementary Fig. S16 and the added sentence reads: “In e-f, At 750 copies per phage, B4GalT1 converts S1 to P1 in 1 day. The starting material peak S1 was consumed by 1 day and any minor residual amount of S1 observed at day 1 did not change after a prolonged reaction (4-5 days). In contrast, significant S1 peak was observed up to 7 days of B4GalT1 modification at 1000 copies per phage.”
2. We added clarification to Supplementary Fig. S17 and the added sentence reads: “In e-f, At 750 copies per phage, Pd26ST converts P1 to P2 in 3 days. In contrast, P1 peak was observed up to 4 days at 1000 copies per phage.”
3. We also added the following paragraph to the results of main text and included the recommended reference:
“Modification with both B4GalT1 and Pd26ST did not proceed as efficiently for densely glycosylated phages (1000 glycans per phage) as they did for phages with medium density glycosylation (750 or less). Specifically, galactosylation of phage containing 1000 copies of glycan by B4GalT1 required seven days for completion, whereas modification of 750 glycans/phage by the same enzyme was done in one day (Supplementary Fig. S16). Similarly, modification by Pd26ST was slower at 1000 glycans/phage than 750 or lower densities (Supplementary Fig. S17). Similar trends were observed for modification by Pd26ST.

(Supplementary Fig. S17) Such observations resemble observations from Fitch, Webb, and co-workers that demonstrated a dramatic reduction of the rate of enzymatic modification of glycans on high vs. low density liposomes⁵⁸. The combined results confirmed that multi-step enzymatic glycan remodeling can be used to create *N*-glycans directly on phage, but that the modification rates slow down near a density of 1000 glycans per 0.7 micron-long phage.”

4. We also added the following paragraph to the discussion of main text “Poor modification of glycans by glycosyltransferases on phages modified by 1000 glycans or 40% occupancy of pVIII proteins (Supplementary Fig. S16-17) is consistent with the poor accessibility of the same 1000 glycan/phage constructs by most lectins; from twelve tested lectins, nine cease any binding when their glycan epitope are displayed on phage at 1000 copies per phage. Interestingly, four lectins: SiaFind™ α -(2→6)-Specific reagent, SiaFind™ Pan-Specific Lectenz[®], CD22, and WGA uniquely increases binding at 1000 copies. Such preference or avoidance of high copy number glycans was confirmed using multi-barcode encoding (Supplementary Fig. S41). Grafting of *N*-glycan X-ray structures to X-ray structure of M13 bacteriophage suggests that packing of glycans at 40% of p8 protein creates dense cylindrical display in which glycan epitopes might not be accessible (Supplementary Fig. S36). We caution that such static grafting should be used with caution as it does capture *N*-glycan dynamics.”

R1.3. Similarly, a discussion summarising the lectin binding results in a broader context- main messages in terms of biological significance - might be useful to a general reader.

We added the following clarification to the discussion of main text “For example, the interplay of glycan density and binding of immunomodulatory proteins such as CD22 and DC-SIGN can be used to understand how cellular density of displayed glycans regulate engagement of these lectins. We observe that binding to DC-SIGN(+) cells requires higher density of paucimannose with GlcNAc capping vs. without capping; and similar observations was made in binding of DC-SIGN cells to CHO glycosylation mutant Lec8 that expresses *N*-glycans⁶⁵. Similar, density-dependent recognition of sialic acids by Siglecs on phage by CD22(+) cells may aid explaining the recognition of low and high densities of sialic acids on cells by Siglec(+) immune cells⁷³.”

Reviewer #2 (Remarks to the Author):

*R2.1. This study addresses two interesting issues in the field of glycobiology, namely how are glycans recognized and what proteins bind to them. To this end, the authors illustrate here a phage-display approach in which glycan presentation is created with what is termed different densities, and an approach is used to tag such glycans on phage in what is termed ‘Genetically-Encoded Multivalent Liquid *N*-glycan Arrays’. While the title is a bit misleading, as discussed below, the principle explored here to generate ‘glycoconjugates’ on phage (restricted to a few Asn-linked or *N*-glycans) at different densities of conjugation and test their binding in vitro to several lectins or carbohydrate-binding proteins, and also inject them into mice and observe the locations of homing for these derivatized phage.*

The work is interesting to a degree, although somewhat preceded by rather similar approaches by this group, and limited to only a few observed glycans or lectins. In essence the study addresses the issue of glycan multivalency and how it affects binding of lectins. The results are not particularly unpredicted, though there are some unusual findings, yet also somewhat inconsistent results within the study. A prior study recently published in Nature Chem. Biol., entitled “Genetically encoded multivalent liquid glycan array displayed on M13 bacteriophage” presents the general approach. The current manuscript does not appreciably extend the observations in novel ways beyond that work. So a major question that this reviewer considered, is that no doubt there is a lot of work to produce the results enclosed herein, but

are the results of these studies mainly predictable based on prior studies in the field? On the whole, most of the conclusions of the study are not particularly novel or unpredictable. There are several conceptual issues that trouble this reviewer. While accepting the fact that this work is both interesting and obviously a lot of effort went into it, and what was done here, was done extremely well, [...]

We thank reviewer for their positive feedback and assessment that all the work in this manuscript is interesting and done extremely well. We address all of the remaining concerns below.

R2.2. *[...] the study falls short in impact and mechanism. To start with, the title of the paper is a bit misleading, in terms of indicating that they have constructed an “N-glycan library” and an “Array” and that it is “Genetically Encoded”.*

The information about the glycan and density are encoded in the genome of the phage. This genome is made of DNA. Encoding can be called genetically-encoded or DNA encoded. We prefer the former to the latter to avoid confusion with DNA-encoded libraries (DEL) in which glycans and small molecules are linked directly to DNA. As detailed below (**R2.3–R2.7**), we feel the use of the terms “N-glycan library” and “Array” are appropriate.

R2.3. *The prior paper cited above generated a liquid glycan array (LiGA) platform, and used a total of 140 glycan-modified phage, while the current paper under review here only used “...a library of six N-glycans”. The argument could be made that this limited number is studied in a different way, i.e. density, from the prior studies*

Question: Should different densities of the same glycan count as different elements in the array?
Answer: Yes, because it is known that changes in density of glycans on the surface of cells,^{20,35-37} glycan arrays arrays,^{19,29,30} polymers, dendrimers, liposomes, and other carriers³¹⁻³⁵ leads to different biological outcomes (citations are from the introduction).

R2.4. *[...] but again, with so few glycans and lack of controls (isomeric glycans as mentioned below), the conclusions are not rigorously supported. The authors only study a total of 6 glycans, hardly a ‘library’ in the literal sense, and these 6 glycans lack diversity, which is hardly impressive.*

As a corollary to **R2.3**, phage that encode different densities of the same glycan should be counted as “different elements of the array”.

Hence, the original manuscript detailed the synthesis of 38 new different glycan-modified phage. Specifically, eight different N-glycans have been presented at different densities. Of these, 35 were tested for lectin binding (Figure. 5). As described in the manuscript, to this mixture, 89 previously published glycan-modified phage were added. The resulting mixture of 89+35=124 glycan-modified phage was injected into mice to understand their homing to different organs (Figure. 6 and Supplementary Fig. S42-48). We are not aware of any work in which glycan arrays have been used to map the organ homing preferences of 124 glycoconjugates inside an animal.

See further information in **R2.7.** and “List of All glycans. xlsx” file was provided as supplementary data file.

R2.5. *First, they are all N-linked, [...]*

The reviewer is correct. It is reflected in the title of our manuscript “Chemoenzymatic Synthesis of Genetically-Encoded Multivalent Liquid N-glycan Arrays”.

However, not all glycans used in this manuscript are N-linked. As noted in **R2.4**, in the animal work, we added phage modified by other classes glycans. These are summarized in Supplementary Fig. S42–S48 and “List of All glycans. xlsx” file was provided as supplementary data file.

R2.6. [...] and no ‘isomers’ of glycans are used, as in the obvious alpha2-6 sialic acid versus alpha2-3 sialic acid modifications. Differential recognition of specific glycan structures that are isomeric in structure would be more appropriate to gauge any level of specificity.

We performed the following experiments to address this question. The legend of each supporting figure is duplicated here as well:

1. We used an α -(2→3)-sialyltransferase (Pm2,3ST) to make α -(2→3)-linked sialo N-glycans as shown in Supplementary Fig. S10 and S11a-c.

Supplementary Figure S10. Model study of Pm2,3ST-catalyzed sialylation on LacNAc-modified pVIII.

a, Scheme of α -(2 →3)-linked sialylation. **b**, Monitoring progress by MALDI-TOF MS. Reaction approached completion by 5 hours of incubation.

Supplementary Figure S11. On-phage enzymatic extension of Gal-terminated N-glycan by Pm2,3ST and structure validation by binding to diCBM40 lectin.

a, Scheme of α -(2 →3)-linked sialylation. Prior to reaction, 8 phage clones with defined SDBs were combined to create a mixture with multiple SDB (MSDB). This MSDB was used to perform the modification by DBCO-NHS, ligation of biantennary N-glycan, and then its enzymatic modification. **b**, Scheme of mixing 8 phage clones to create multiple SDB (MSDB). **c**, MALDI-TOF MS indicated substrate S1 was consumed after 1 h incubation. The negative charge of terminal α -(2 →3)-linked sialic acid affected the product P1 detection by MALDI-TOF MS. The desired product peak P1 was not visible in MALDI-TOF MS. **d**, To validate the formation of terminal α -(2 →3)-linked sialic acid N-glycan on MSDB, we mixed MSDB glycoconjugates with LiGA6×5, fucosylated N-glycans, α -(2 →3)-linked sialic acid O-glycans and fucosylated O-glycans. The mixture was tested for binding to diCBM40. The results confirmed the binding of glycans with terminal α -(2 →3)-linked sialic acid to diCBM40.

2. We tested the binding of isomeric α -(2→3)- vs α -(2→6)-linked sialic acid N-glycans by three lectins and observed differential recognition of isomeric glycans as shown in Supplementary Fig. S11d (see above), and S30–32.

Supplementary Figure S30. Binding of LiGA6×5 to sialic acid-specific diCBM40 lectin before and after remodeling by Pm2,3ST.

a, The LiGA6×5 was treated with Pm2,3ST to transfer CMP-sialic acid onto Gal-terminal N-glycans displayed on phage. After 1 h incubation at 37 °C, PEG/NaCl was added to precipitate the phage. HBS buffer was used to dissolve the phage. Resulting solution, termed “LiGA6×5-2,3ST” was added to diCBM40 coated well to validate the binding of newly introduced α -(2→3)-linked sialic acids to diCBM40 **b**, Structures displayed on phage of LiGA6×5. **c**, Binding of unmodified LiGA6×5 to diCBM40: α -(2→6)-sialosides exhibited enrichment by diCBM40,

because diCBM40 can recognize α -2,6-sialosides with weak affinity¹⁶. **d**, Structures displayed on phage in LiGA6 \times 5-2,3ST. **e**, Binding of modified LiGA6 \times 5-2,3ST to diCBM40: we observed strong binding of diCBM40 towards α -(2 \rightarrow 3)-sialosides on phage that contained only terminal Gal prior to modification (red rectangle). In a set of isomeric glycans, pure α -(2 \rightarrow 6) (yellow) binds weaker than pure α -(2 \rightarrow 3) (red) and mix of α -(2 \rightarrow 3) and α -(2 \rightarrow 6)-sialosides (green). Increased binding to α -(2 \rightarrow 3) versus α -(2 \rightarrow 6)-sialosides is consistent with known preferences of diCBM40 for α -(2 \rightarrow 3) linkage.

Supplementary Figure S31. Binding of LiGA6 \times 5 to SiaFindTM Pan-specific Lectenz[®] before and after remodeling by Pm2,3ST.

a, The remodeling procedure as described in Supplementary Figure 30. **b**, Structures displayed on phage as LiGA6 \times 5. **c**, When there is only α -(2 \rightarrow 6)-sialosides of LiGA6 \times 5, we observed binding to Pan-specific Lectenz. **d**, Structures displayed on phage of LiGA6 \times 5-2,3ST. **e**, We observed binding of Pan-specific Lectenz towards both to α -(2 \rightarrow 6)-sialosides and newly installed α -(2 \rightarrow 3)-sialosides.

Supplementary Figure S32. Binding of LiGA6 \times 5 to SiaFindTM α -(2 \rightarrow 6)-specific reagent before and after remodeling by Pm2,3ST.

a, The remodeling procedure was describe in Supplementary Figure S30. **b**, Structures displayed on phage as LiGA6 \times 5. **c**, We observed binding of α -(2 \rightarrow 6)-sialosides to SiaFindTM α -(2 \rightarrow 6)-specific reagent. **d**, Structures displayed on phage of LiGA6 \times 5-2,3ST. **e**, Unlike SiaFindTM Pan-specific Lectenz[®] (Supplementary Figure S31) or diCBM40 (Supplementary Figure S30), SiaFindTM α -(2 \rightarrow 6)-specific reagent recognized only α -(2 \rightarrow 6)-sialosides but newly installed α -(2 \rightarrow 3)-sialosides did not show binding.

We also added the following text to the main text:

“We further demonstrated that α -(2 \rightarrow 3)-sialyltransferase (Pm2,3ST) can be used on model LacNAc glycans displayed on phage (Supplementary Fig. S10) and Gal-terminated N-glycans displayed on phage (Supplementary Fig. S11a-b).”

“To provide additional support for this affinity hypothesis, we tested a dimeric SiaFindTM α -(2 \rightarrow 6)-specific protein, which binds to α -(2 \rightarrow 6)-sialic acid with micromolar K_d and monomeric SiaFindTM Pan-specific Lectenz[®], which binds both α -(2 \rightarrow 6) and α -(2 \rightarrow 3)-sialic acid with 0.2–0.4 μ M affinity. Unlike tetrameric SNA lectin, both SiaFindTM reagents preferred α -(2 \rightarrow 6)-sialoglycans at higher density (Supplementary Fig. S31c and S32c). This experiment suggests that density-dependence might emanate from oligomerization state rather than affinity.”

“Enzymes can be used to remodel not only individual glycans but also their mixtures (libraries). To show this capacity, we remodeled LiGA6 \times 5 using α -(2 \rightarrow 3)-sialyltransferase (Pm2,3ST) and showed that the glycans in the remodeled library bind to diCBM40 (Supplementary Fig. S30e), SiaFindTM Pan-specific Lectenz[®] and SiaFindTM α -(2 \rightarrow 6)-specific reagent (Supplementary Fig. S31e and 32e). We observed that SiaFindTM Pan-specific Lectenz[®] binds to the newly installed α -(2 \rightarrow 3)-sialosides but SiaFindTM α -(2 \rightarrow 6)-specific reagent does not. Again, SiaFindTM Pan-specific Lectenz[®] preferred α -(2 \rightarrow 3)-sialosides only at high density. SiaFindTM Pan-specific Lectenz[®] and SiaFindTM α -(2 \rightarrow 6)-specific reagent are monomeric and dimeric, respectively, whereas SNA is tetrameric and the observed density preferences might stem from oligomerization state of lectin rather than K_d which is approximately micromolar for all.”

We believe that these experiments address both concerns.

R2.7. More glycans would also be helpful to make an argument about a library.

To address the concerns of this reviewer, we created 25 new glycoconjugates in addition to the 124 glycoconjugates already used in the original manuscript. The new version of this manuscript contains a total of 149 glycoconjugates. In our opinion, 149 compounds is a library (See Supplementary Data for the list of all glycans used in this manuscript).

R2.8. Much of the work describes the enzymatic modification of glycans on the phage, and while somewhat interesting, the results are rather predictable. For example, galactosyltransferase is shown to modify a terminal GlcNAc-containing N-glycan (Fig. 5), partly and then to completion. Such reactions as shown in this manuscript have been used by many laboratories for many years, so the preparation of the modified glycans and most of the information about this lacks novelty. Similar predictability applies to removal of glycans, as the observation that neuraminidase can completely remove sialic acid, which would be expected, of course (Fig. 2).

The following text was added to the discussion: “To the best of our knowledge, enzymatic remodeling of glycans on bacteriophage surfaces have not been reported. Enzymatic remodeling of glycans in solution has been used by many laboratories^{71,72} but applying such reactions to remodeling of immobilized glycans is not trivial. Enzymatic modification of immobile glycans on solid surfaces often show slower enzymatic conversion when compared to analogous conditions in solution^{71,72}. Webb, Flitch, and co-workers demonstrated that modification of mobile glycans on the surface of liposomes also proceeds significantly slower than modification in solution⁵⁸. The authors elegantly demonstrated that increased density of glycan acceptor on liposome further reduced the rate of modification suggesting that at low density, the decrease may arise from mass-transport limitations of the diffusion of enzyme to this 2D surface of 100–1000 nm liposome whereas steric occlusion plays additional role in liposomes with high density of glycan acceptor⁵⁸. LiGA displays immobile glycans on a 5–700 nm cylinder (dimensions of M13 bacteriophage) and the enzymatic modification on phage mirror observations from synthesis of glycans on liposomes.”

R2.9. The concept of density control is not strictly accurate to this reviewer, as the illustrations indicate, the density is not uniform, but randomly varied and a lot of different densities are present as a result of the extent of modification. Such site density effects have been explored with neoglycoproteins in the past made on bovine serum albumin with random degrees of modification, and similar arguments or qualifications can be made for those. The better term might be degree of modification (500, 750 or 1000 glycans) (see Fig. 9).

Density is a canonical description of an ensemble property. It is defined as an average number of entities per unit volume or unit surface. We believe this reviewer is confusing the bulk property (density of glycans) with local property (distance between the glycans). We did not claim there was uniform distances between the glycans. Also, we prefer to not develop a different term for “density”. Nevertheless, to avoid confusion, we added the following text to our discussion:

“In this paper we use the term “density” to denote the degree of modification of pVIII proteins by glycans (500 out of 2700 available, 750 out of 2700 available, *etc.*). While the distances between the pVIII proteins are rather uniform, the distances between the glycans fluctuated

around the average, with some local regions where the glycans are closer and further apart than on average (Supplementary Fig. S36). There are only a few multivalent scaffolds in which such fluctuations are augmented: notable exceptions are short rigid DNA molecules that carry glycans on two ends⁷⁶, well-packed and defect-free self-assembled monolayers of glycans on gold⁷⁷; bacteriophages in which most capsid protein has been modified by a glycan²⁰. However, fluctuations of glycan-to-glycan distances are unavoidable for multivalent scaffolds in general either due to stochastic substitution or flexibility of the scaffold⁷⁸. Such scaffolds in principle mimic lectin–glycan interaction in physiological conditions in the presence of such the local variations in distances between the glycans on the cell surface²⁴.”

R2.10. The paper seems more of a methods development, but not especially novel in that case either, as the prior published paper basically provided that technology. The simple array itself is only genetically-encoded in that phage DNA is encoded, not the glycan, and the entire technology of employing phage virions with DNA barcodes in phage genome was described.

We disagree that “phage DNA is encoded not the glycans”. Glycans are DNA encoded in LiGA.

We ligate glycans to a prospectively DNA barcoded virion. There is 1:1 correspondence between the glycan structure or its presentation (density) and the DNA barcode inside the phage. This is the definition of DNA encoding.

We believe that perhaps the reviewer is confusing “encoding” with “templated synthesis”. Indeed, glycans are not constructed using DNA as a template. We are not claiming that the synthesis of glycans is templated by DNA or guided by DNA in any fashion. Still, they are encoded by DNA in LiGA. We hope our reply resolves any confusion.

We also disagree about the lack of novelty, which is addressed in detail below (**R2.14**)

R2.11. A lot of the paper used prior methods and lacks originality. The method of preparing the Multivalent Liquid N-glycan Array is not novel, as it is created by the same approach as used previously using alkyl-azido linkers to dibenzocyclooctyne (DBCO)-modified M13 phage. The method of remodeling of the egg-derived sialoglycopeptide is by conventional techniques.

The reviewer is correct, we are not inventing methods for attachment of glycans because the DBCO approach works. It is convenient for ligation. We do uncover a new observation in which the rate of ligation of the same N-glycan to DBCO on phage decreases as the copy number of DBCO on phage increases (Supplementary Fig. S15). Additional comments about novelty are found in **R2.14**.

R2.12. In regard to what this reviewer considers the two major discoveries of this work, is that at extremely high conjugation levels some lectins did not bind the phage. The explanation for this is not really provided, other than “presumably due to steric occlusion of tightly packed epitopes”, which is not really an explanation, but a restating of the observation.

We performed the following experiments as shown in Supplementary Fig. S35–36. to address this question and the legend of each figure is duplicated here as well:

Supplementary Figure S35. MALDI-TOF MS results of five phage conjugates for the study of steric occlusions.

a, Five different phage clones that contain different ratios of α -(2 \rightarrow 6)-sialylated and galactose-terminated N-glycans on the same phage virion. **b**, 1000 copy numbers of five DBCO modified

phage clones were reacted with different ratio of α -(2 →6)-sialylated and galactose-terminated N-glycans. **c**, Expanded figure of MALDI-TOF MS analysis from **b**.

Supplementary Figure S36. Steric occlusions in the interaction between lectins and N-glycans on phage.

a, To test the steric occlusion hypothesis, we produced LiGA-SO in which phage displayed either different densities of N-glycans **9** and **10** or mixtures of glycans **9** and **10** on the same phage. In the latter mixture, the total number of glycans remained 1000 glycans per phage but the ratio between **9** and **10** varied (see Supplementary Figure S35). **b**, Binding of the LiGA-SO to ECL, SNA and WGA demonstrated that binding of all lectins ceased when total density of glycans reached 1000 copies. Specifically, 500 copies of glycan **9** on phage exhibited strong binding to ECL and WGA, 500 copies of glycan **10** exhibited only minor binding to the same lectins but when 500 copies of **9** and 500 copies of **10** were present on the same phage simultaneously, the binding was abrogated. Also, within the populations of phage that displayed 1000 copies of **9** and **10**, the strongest binding was for phage that displayed 750:250 ratio of **9** and **10** (Supplementary Fig. S36b). Introduction of non-binding glycans on phage decreases the access to the productive glycans. This experiment confirmed that steric occlusions played a role in the decreased binding of lectins to high density of glycan-modified phages.

We also added the following text to the main text:

“We previously confirmed that decrease of lectin–glycan interactions at high glycan density occurs due to steric occlusion^{38,61}, here we tested this again by making a phage that displayed a mixture of sialylated and non-sialylated glycans (Supplementary Fig. S35). The mixtures of functional and non-functional glycans did not bind to SNA when the total density of glycans reached 1000, even when density of sialoglycans was low in such mixture (Supplementary Fig. S36).”

“In agreement with our previous report, the binding to Man₃ was ablated at ≥1000 glycans/phage due to steric occlusion of tightly packed Man₃ epitopes³⁸.”

“We tested whether binding to ECL and WGA at high densities of glycans decreased due to steric occlusion (Supplementary Fig. S36). Specifically mixing binding glycan **10** and weakly-binding glycan **9** on the same phage yielded a multivalent display that was only weakly binding.”

R2.13. The second major observation arises from the experiments in which LiGA are injected into the mouse tail vein, and phage levels are determined in recovered plasma and organs after 1 hour. The authors found that many modified phage accumulated in spleen, which is attributed, but not shown, to be due to the abundance of endogenous lectins in lymphocytes, and other accumulated in liver, presumably due to the liver asialoglycoproteins receptor (ASGPR), but this is also not proven.

We performed an experiment and added new clarification to the main text: “We observed that phage clones displaying P1 tetra, Globoside P, GD2 also bound to HepG2 cells, which express ASGPR receptors; the binding was 10–20x higher (p<0.001) when compared to unmodified phage (Supplementary Fig. S49).”

As we cannot attribute any glycan binding to any specific lectin in the spleen, we also changed the wording in sentence related to spleen homing. The resulting sentence reads: “The significant (FDR<0.05) spleen enrichment of diverse N-glycans (Fig. 6b) and synthetic glycans (Supplementary Fig. S42) may be attributed to binding to lectins expressed by lymphocytes, macrophages, dendritic cells, and plasma cells residing in spleen.”

R2.14. *display platform with glycans has already been published, as in the cited recently developed liquid glycan arrays (LiGAs) (reference 39).*

Below we summarize the information in this manuscript which is not present in already published report nor in any reports published to date:

1. Complex N-glycans are displayed on phage at a controlled density ranging from 50 to 1000 copies and investigated *in vitro*, on cells and in live mice *in vivo*.
 - This has never been performed before.
2. Diversification of glycans on phage by enzymatic trimming and multi-step extension.
 - This has never been performed on any bacteriophage and not with DNA encoding.
3. Density-dependence of chemoenzymatic transformations on phage.
 - There is one similar precedent on liposomes⁵⁸.
4. Remodeling of entire libraries by enzymatic diversification.
 - This is, in principle, feasible on other arrays^{47,48} but rarely employed due to incomplete conversions and lack of characterization.
5. First *in vivo* investigation of the homing preferences of >100 diverse glycans in live animals.
 - This has never been performed before.
6. Avidity response of N-glycans across 15 glycan binding proteins and cell surface receptors using one integrated technology.
 - This has never been performed before.

R2.15. *The identification of glycans recognized in this approach using in vitro approaches was limited to 10 lectins. No studies were performed on the endogenous lectins in mice that might be responsible for homing of conjugates.*

The study is no longer limited to ten lectins. There are now 10+5 glycan-binding proteins investigated (new lectins are: SiaFind™ α -(2→6)-Specific reagent, SiaFind™ Pan-Specific Lectenz®, diCBM40, AAL, and ASGPR). One of them was ASGPR expressed on the surface of HepG2 cells (see reply to comment **R2.13**. The latter receptor responsible for homing of GalNAc terminated glycans to liver.)

R2.16. *The artificial approach of presenting clustered glycans on a phage-display approach may not represent any biological situation, and indeed the authors do not identify any specific or known interactions that represent displays of density versus binding that they encounter with this phage display method.*

With this comment the reviewer effectively discards the usefulness of any “*artificial approach of presenting clustered glycans*” on glass arrays, beads, dendrimers, liposomes, polymers, etc. We find this surprising given the large number of novel insights that the larger glycobiology community has obtained using approaches of this sort over the past 40–50 years. The second paragraph of the introduction in both the earlier and current version of the manuscript have a caveat that recognizes the limitations of arrays vis-à-vis the ‘real’ world. We feel that is sufficient to point out this issue, which is widely appreciated and, in our opinion, does not require further elaboration.

R2.17. *The authors make a misleading statement that “The affinity between monovalent glycans and a GBP binding site is typically low”. This type of statement is often made, but applies to non-biological or small oligosaccharides, as binding of GBPs to natural glycan ligands are typically relatively high affinity*

in the nm to low micromolar range. Of course, by clustering poor ligands, one can increase avidity, but whether this is truly biological representative is not well documented. It seems to be rather a belief. In any case, the authors do not actually describe affinity constants of binding, so this aspect of the work is lacking.

The misunderstanding emanates from the usage of qualitative adjective “low”. To resolve it, we measured K_d of six N-glycans to six lectins using ESI-MS. The obtained the K_d values of all N-glycan: lectin pairs are now available in the Supplementary Fig. S33. The 36 K_d s ranged from 300, 500 and 700 nM to low micromolar and millimolar range.

We added the following clarification to the main text: “The affinity between monovalent glycans and a GBP binding site are typically in the low μ M range.”

We also referenced Supplementary Fig. S33. in the following sentence of main text: “[..] 50–100x weaker affinity of interaction of sialylated glycans with CD22 when compared to SNA (75 μ M and 0.77 μ M respectively)^{61,62} (Supplementary Fig. S33-34).”

Reviewer #3 (Remarks to the Author):

R3.1. *Glycan-protein interactions play important roles in many biological processes, but methods to study them have key limitations. One of the most common and powerful approaches is glycan array technology. Standard arrays typically contain many different glycans immobilized on a glass slide. Some alternatives include bead-based arrays and cell-based arrays. The Derda group has pioneered a new format – glycan libraries on genetically encoded phage particles. It’s an excited new approach that allows analysis of samples that are not compatible with solid supports like glass. For example, a phage library can be injected into an animal to evaluate localization of glycans to many different parts of the body in parallel.*

In this paper, the authors describe an advance to their phage array approach. A key challenge is getting glycans for the array. In this study, they address this limitation by developing methods to attach one or more glycans to the phage and then diversify that library on phage by enzymatically extending and/or trimming the glycans. It’s an efficient, versatile strategy. One really nice aspect of their study is their ability to follow enzymatic reactions by mass spectrometry. It can be quite difficult to track reaction outcomes/progress with large particles, but it works really well for their system. The ability to monitor progress allowed them to optimize the reactions in ways that are normally quite difficult. As a result, they were able to develop conditions to get essentially complete reactions. This is impressive because enzymatic glycan extension and trimming can often be incomplete, leaving mixtures. As an added layer of information, they also investigate effects of glycan density on the phage. In the end, they built a library of 6 N-linked glycans at 5 different densities for 30 combinations. They then used it to profile a variety of lectins. Lastly, they injected them into mice to track organ localization – something that is impossible with other standard array formats.

Overall, it is a pretty interesting study. After clarifying a few things (see below), it will be a nice addition to Nature Comm.

We thank reviewer for the very positive feedback and have addressed each of the specific comments below.

R3.2. I may have missed it, but how did they authors account for multiple comparisons when doing their statistical analyses? Perhaps add some info in the supporting information.

In the caption of Fig. 5 we now state:

“multiple test control was by using false discovery rates (FDR) from Benjamini-Hochberg adjustment,” and in the Fig. 6 caption, “multi-test control by FDR from B-H correction.”

In the SI Section 1.1 General data processing methods the text has been modified: “...the abundances between two sets of samples can be tested for significant differences, in order to deal with the many such tests, Benjamini–Hochberg (BH) adjustment is used to control the false discovery rate (FDR) at $\alpha = 0.05$.”

R3.3. Figure 6. the legend lists b twice.

The text “heatmap b” preceded a parenthesis, and appeared to be a second subpanel b label. We have altered the caption to avoid this confusion. The changed sentence reads: “c, Focusing only on the liver data in the heatmap, we summarize enrichment of the four classes of N-glycans, n=3 data points represent data from n=3 animals.”

R3.4. For the second listing, I don't really understand the Mann Whitney part – what part of the figure are they referring to and which things are significantly different? Should there be asterisks and brackets somewhere?

We have redrawn Fig. 6c to show the p-value of 0.02 on a bracket between the Gal-terminal glycans and the other cases. We have not tried to add the bracket to the plot because there are three different regions with the Gal-terminal glycans. We have also altered the caption text.

R3.5. In part d, having 6 bar graphs and 1 p value is a little confusing. what are the specific differences that are significant?

We have redrawn Fig. 6d to show the p-values on a bracket between the hepatic samples and the other cases.

R3.6. The phage array SAR for SNA looks pretty surprising. Their data suggest that a terminal galactose on one arm plays a key role in recognition. Its absence or extension significantly hampers binding. If the authors think this is real, I would suggest mentioning it in the main text with some discussion, ie. is this known or new, and why would it have a big effect. If they think it is an artifact or just variability of the measurement, I would mention that in the supporting information.

We analyzed the statistic significance of this observation and saw that the difference between N-glycans 7, 8 and 9 in Fig. 5 is not significant. To investigate the differences in binding of SNA to N-glycans 7, 8 and 9, we measured the monovalent affinities of these glycans to SNA (Supplementary Fig. S33). Binding to glycan 9 was the weakest (0.8 ± 0.2) and binding to 8 was the strongest (0.3 ± 0.1), but we observed insignificant difference between binding of glycan 7 and 8 to SNA. The minor observed fluctuations in binding might also result from differences in accessibility of glycans 7 and 8 in a multivalent display on phage.

R3.7. Some of the ConA data is surprising too. The density effect for ConA with Man₃ is a lot larger than I would have expected. Also, the lack of binding to Man₃GlcNAc₂ is surprising given the Cummings and CFG data. Any explanation for these effects? a little more discussion in the main paper could be helpful here. Are there key differences in the platforms leading to these differences in binding profiles?

We performed the following experiments as shown in Supplementary Fig. S37–39 to address this question and the legend of each figure is duplicated here as well:

Supplementary Figure S37. Comparison of LiGA-mixtures with and without binding paucimannose N-glycans.

a, We created LiGA₂×₆ which contained paucimannose biantennary N-glycans on phage with terminal GlcNAc (glycan **6**) and paucimannose alone (glycan **11**). The LiGA₂×₆ was screened against ConA. Only Man-terminated conjugates were enriched. **b,** To test whether presence of strong binder interferes with binding of weaker binder **6** on phage, we made another LiGA mixture without paucimannose. Glycan **6** on phage did not bind to ConA even when glycan **11** on phage was not present. We tested two different ratios between phage displaying glycan **6** and control phage. Both results showed no binding.

Supplementary Figure S38. The workflow of creating MSDB glycoconjugates.

a-d, Each set of MSDB consisted of 7 phage clone with a unique DNA barcode and conjugated individually to DBCO-NHS. Low to high densities of DBCO-MSDB phage were created and conjugated to biantennary N-glycan **6**. After MALDI showed full conversion of conjugation to generate peak S1, β-N-acetylglucosaminidase S was added to cleave terminal GlcNAc yielding low to high densities of MSDB paucimannose-conjugates P1. These individual MSDB glycoconjugates were mixed with LiGA₆×₅ for binding experiments as shown in Supplementary Figure 39-41.

Supplementary Figure S39. Binding of ConA to MSDB-paucimannose conjugates and LiGA₆×₅.

a, The remodeling procedure was described in Supplementary Figure S30. **b,** We used the same mixing procedure as described in Supplementary Figure S38. then measured binding with ConA. **c,** This MSDB-Man₃ scan showed that binding of ConA to phage displaying 150 glycans is better than 50 or 500 and binding to any construct with 50–750 glycans per phage are better than binding to phage displaying 1000 glycans. Also, we again observed the phage clones that displayed **6** showed no binding, which is consistent with the results shown in Supplementary Figure 37. **d,** Structures displayed on phage of LiGA₆×₅- β-N-acetylglucosaminidase S. **e,** When the LiGA was treated with β-N-acetylglucosaminidase S, the non-binding clones that used to carry glycan **6** exhibited strong binding, due to removal of GlcNAc.

We also added the following clarification to the main text:

“To test the significance of density-dependent binding, we produced a library in which every density of paucimannose was linked to seven distinct SDB (DNA barcodes) (Supplementary Fig. S38). This multi-SDB (MSDB encoding of five densities of Man₃) confirmed that phage displaying 150 paucimannose N-glycans bound to ConA significantly better than phage that display 50 or 500 copies of the same glycan. Furthermore, binding to 1000 copies of paucimannose per phage is significantly weaker than any other construct (Supplementary Fig. S39).”

Lack of ConA binding to paucimannose “with terminal GlcNAc is not in agreement with earlier observations by Cummings and coworkers (Supplementary Fig. S23c)²⁸ and the Consortium for Functional Glycomics (CFG) array (Supplementary Fig. S23e). We observed no binding at any glycan density (Supplementary Fig. S37a) and to rule out the possibility of competition between

glycans, we prepared simple mixtures (Supplementary Fig. S37b) that contain only paucimannose with GlcNAc **6**. Again, the phage clones that displayed **6** showed no binding.”

“However, when the LiGA was treated with β -N-acetylglucosaminidase S, the non-binding clones that used to carry glycan **6** exhibited strong binding, due to removal of GlcNAc (Supplementary Fig. S39e) while binding of other glycans to ConA was not affected. This experiment showed that enzymes can be used to remodel not only individual glycans but also their mixtures (libraries).

R3.8. The GNA binding profiles are also pretty unusual. I find it very surprising the binding would be observed for Man3 at 150/phage and 750/phage but not 500/phage. Any explanation? Is this an experimental issue/artifact? Maybe some sort of barcode issue or processing issue? Perhaps some independent verification- can you do phage ELISA with these to confirm the results?

We performed the following experiments as shown in Supplementary Fig. S40 to address this question and the legend is duplicated here as well:

Supplementary Figure S40. Binding of GNL to MSDB-phage paucimannose conjugates and LiGA6 \times 5.

a, The remodeling procedure was describe in Supplementary Figure S30. **b**, We mixed 5 densities of MSDB-phage paucimannose conjugates with LiGA6 \times 5, and measured binding with GNL. Azidoethanol phage were used as control. **c**, Detailed study using MSDB-Man3 showed that GNL preferred binding to paucimannose displayed on phage with intermediate density (150-750 copies).

We also added this to the main text:

Finally, the GNL lectin, which is known to recognize paucimannose, “bound strongly to phages that display intermediate copy number of this structure (150–750 copies, Supplementary Fig. S26b) and more weakly to any other *N*-glycans (Fig. 5h, Supplementary Fig. S26). A detailed study of GNL:Man₃ recognition using MSDB-Man₃ library described above confirmed this observation (Supplementary Fig. S40).”

R3.9. The WGA binding to Man3 is pretty surprising too. It seems unusual to have good binding at a density of 750/phage but not 500 or 780. I'm also surprised that it is one of the best binders at 5 and 1ug/mL. Is that in line with prior published data for WGA?

We performed the following experiments (Supplementary Fig. S41) to address this question and the legend is duplicated here as well:

Supplementary Figure S41. Binding of WGA to MSDB-paucimannose conjugates and LiGA6 \times 5.

a, We used the same procedure as described in Supplementary Figure S38 then measured binding with WGA. **b-c**, We tested the significance of density dependent binding of WGA to paucimannose on phage by using MSDB-Man₃. We observed binding of WGA to MSDB-Man₃ at densities between 500 to 1000 glycans per phage. No binding occurred at densities of 50 and 150 copies of paucimannose.

We also added the following clarification to the main text:

“We further tested the significance of density dependent binding of WGA to Man₃ by using MSDB-Man₃ (Supplementary Fig. S41). We observed binding of WGA to MSDB-Man₃ at densities from 500 to 1000 glycans per phage. No binding occurred at densities of 50 and 150 copies of paucimannose.”

R3.10. In the liver, spleen and kidney data, the differences between 6'SL at 650 vs 760/phage are pretty surprising. Are these the same glycan at slightly different densities, or am I reading this wrong? Assuming they are the same glycan, why would a density difference of 650 vs 760 have such a large effect? Is it possible that the differences are related to some other aspect of the experiment – like processing side?

We would like to thank the reviewer for pointing out this anomaly in the data. The reviewer is correct, these are the same glycan at two different display densities. In our data, the 6'SL-[650] is considerably enriched in liver, spleen and kidney because of the relatively low read numbers in the plasma samples.

Plasma (reads)

Reads for 6'SL-[650]: [0 0 0 0 1]

Reads for 6'SL-[760]: [38 58 32 64 71 23]

Liver (reads)

Reads for 6'SL-[650]: [5 2 0 0 3 0]

Reads for 6'SL-[760]: [17 18 52 86 95 19]

Spleen (reads)

Reads for 6'SL-[650]: [0 169 341 405 722]

Reads for 6'SL-[760]: [2 51 49 52 132]

Kidney left (reads)

Reads for 6'SL-[650]: [11 10 0 0 1 1]

Reads for 6'SL-[760]: [0 1 0 0 0 0]

Kidney right (reads)

Reads for 6'SL-[650]: [0 0 4 0 3 0]

Reads for 6'SL-[760]: [41 0 0 5 2 0]

We have re-evaluated our data and now mark where the amount of phage in plasma samples was below detection level in Supplementary data and added a clarifying statement in the affected figure captions. The statement reads: “The 6'SL-[650] is considerably enriched higher in liver, spleen and kidney compared to 6'SL-[760] (Supplementary Fig. S42-45) because of the relative low reads in the plasma samples.”

REVIEWERS' COMMENTS

Reviewer #1 (Remarks to the Author):

This manuscript describes extensive studies on a very elegant glycan display system that has the unique characteristics of being able to work both in vitro and in vivo. Following on the first demonstration of the system with a small number of probes, this paper dramatically increases the number of glycans and GBP tested and in particular the experiments in mice are very unique and interesting. The authors have addressed referees comments in full and I am happy to recommend publication of this important paper.

Reviewer #3 (Remarks to the Author):

the authors have addressed my concerns